# Ubiquitous atmospheric production of organic acids mediated by cloud droplets

B. Franco[1,2,13 ✉], T. Blumenstock[3], C. Cho[1], L. Clarisse[2], C. Clerbaux[2,4], P.-F. Coheur[2], M. De Mazière[5], I. De Smedt[5], H.-P. Dorn[1], T. Emmerichs[1], H. Fuchs[1], G. Gkatzelis[1], D. W. T. Griffith[6], S. Gromov[7,8], J. W. Hannigan[9], F. Hase[3], T. Hohaus[1], N. Jones[6], A. Kerkweg[1], A. Kiendler-Scharr[1], E. Lutsch[10], E. Mahieu[11], A. Novelli[1], I. Ortega[9], C. Paton-Walsh[6], M. Pommier[4,12], A. Pozzer[7], D. Reimer[1], S. Rosanka[1], R. Sander[7], M. Schneider[3], K. Strong[10], R. Tillmann[1], M. Van Roozendael[5], L. Vereecken[1], C. Vigouroux[5], A. Wahner[1] & D. Taraborrelli[1,13 ✉]

Atmospheric acidity is increasingly determined by carbon dioxide and organic acids[1–3]. Among the latter, formic acid facilitates the nucleation of cloud droplets[4] and contributes to the acidity of clouds and rainwater[1,5]. At present, chemistry–climate models greatly underestimate the atmospheric burden of formic acid, because key processes related to its sources and sinks remain poorly understood[2,6–9]. Here we present atmospheric chamber experiments that show that formaldehyde is efficiently converted to gaseous formic acid via a multiphase pathway that involves its hydrated form, methanediol. In warm cloud droplets, methanediol undergoes fast outgassing but slow dehydration. Using a chemistry–climate model, we estimate that the gas-phase oxidation of methanediol produces up to four times more formic acid than all other known chemical sources combined. Our findings reconcile model predictions and measurements of formic acid abundance. The additional formic acid burden increases atmospheric acidity by reducing the pH of clouds and rainwater by up to 0.3. The diol mechanism presented here probably applies to other aldehydes and may help to explain the high atmospheric levels of other organic acids that affect aerosol growth and cloud evolution.

Chemical production is estimated to be the dominant atmospheric source of formic acid (HCOOH), with a substantial contribution ascribed to sunlight-induced degradation of volatile organic compounds (VOCs) emitted by plants[6,8,9]. Direct HCOOH emissions are thought to account for less than 15% of the total production[6,8,9]. The overall atmospheric lifetime of HCOOH is 2–4 days, owing to efficient wet and dry deposition in the atmospheric boundary layer[6,7,10], but increases to about 25 days in cloud-free tropospheric conditions.

Here we use the global chemistry–climate model ECHAM5/MESSy[11] (EMAC) to simulate atmospheric HCOOH abundance. The reference simulation (EMAC$_{(base)}$) implements the chemical formation pathways that are usually accounted for[8,9,12] (Methods). Using Infrared Atmospheric Sounding Interferometer (IASI)/Metop-A satellite column measurements[13] to determine the HCOOH burden (Methods), EMAC$_{(base)}$ illustrates the issue (Fig. 1a, b): the model globally underpredicts the satellite columns by a factor of 2–5. Similar biases relative to ground-based Fourier transform infrared (FTIR) columns are observed at several latitudes (Extended Data Fig. 1). These persistent discrepancies point to substantial unidentified sources of atmospheric HCOOH.

Recent studies have proposed several missing sources to explain the model underprediction. These include locally enhanced emissions of HCOOH and its precursors, and updated or tentative chemical pathways that involve a broad range of precursors, primarily of biogenic origin[6,9,12,14]. To match the observed concentrations, the required increase in emissions of the known HCOOH precursors and/or HCOOH yields from hydrocarbon oxidation is inconsistent with our understanding of the reactive carbon budget[7,8,15]. Furthermore, such attempts do not account for the elevated HCOOH concentrations observed in free-tropospheric, low-VOC air masses[13,16,17]. Owing to a lack of supporting laboratory measurements, the proposed chemical pathways are often affected by large uncertainties or are speculative. Currently, no atmospheric model offers a consistent picture of tropospheric organic acids.

Here we present a large, ubiquitous chemical source of HCOOH from a multiphase pathway (Fig. 2). In cloud water, formaldehyde (HCHO)—the most abundant aldehyde in the atmosphere—is a known source of HCOOH in remote regions[5,10,18], via rapid oxidation of its monohydrated form, methanediol (HOCH$_2$OH). Nevertheless, most of the HCOOH

[1]Institute of Energy and Climate Research, IEK-8: Troposphere, Forschungszentrum Jülich, Jülich, Germany. [2]Université libre de Bruxelles (ULB), Spectroscopy, Quantum Chemistry and Atmospheric Remote Sensing, Brussels, Belgium. [3]Institute of Meteorology and Climate Research, Karlsruhe Institute of Technology, Karlsruhe, Germany. [4]LATMOS/IPSL, Sorbonne Université, UVSQ, CNRS, Paris, France. [5]Royal Belgian Institute for Space Aeronomy, Brussels, Belgium. [6]Centre for Atmospheric Chemistry, School of Earth Atmospheric and Life Sciences, University of Wollongong, Wollongong, New South Wales, Australia. [7]Max Planck Institute for Chemistry, Mainz, Germany. [8]Institute of Global Climate and Ecology (Roshydromet and RAS), Moscow, Russia. [9]National Center for Atmospheric Research, Boulder, CO, USA. [10]Department of Physics, University of Toronto, Toronto, Ontario, Canada. [11]Institute of Astrophysics and Geophysics, University of Liège, Liège, Belgium. [12]Ricardo Energy and Environment, Harwell, UK. [13]These authors contributed equally: B. Franco, D. Taraborrelli. ✉e-mail: bfranco@ulb.ac.be; d.taraborrelli@fz-juelich.de

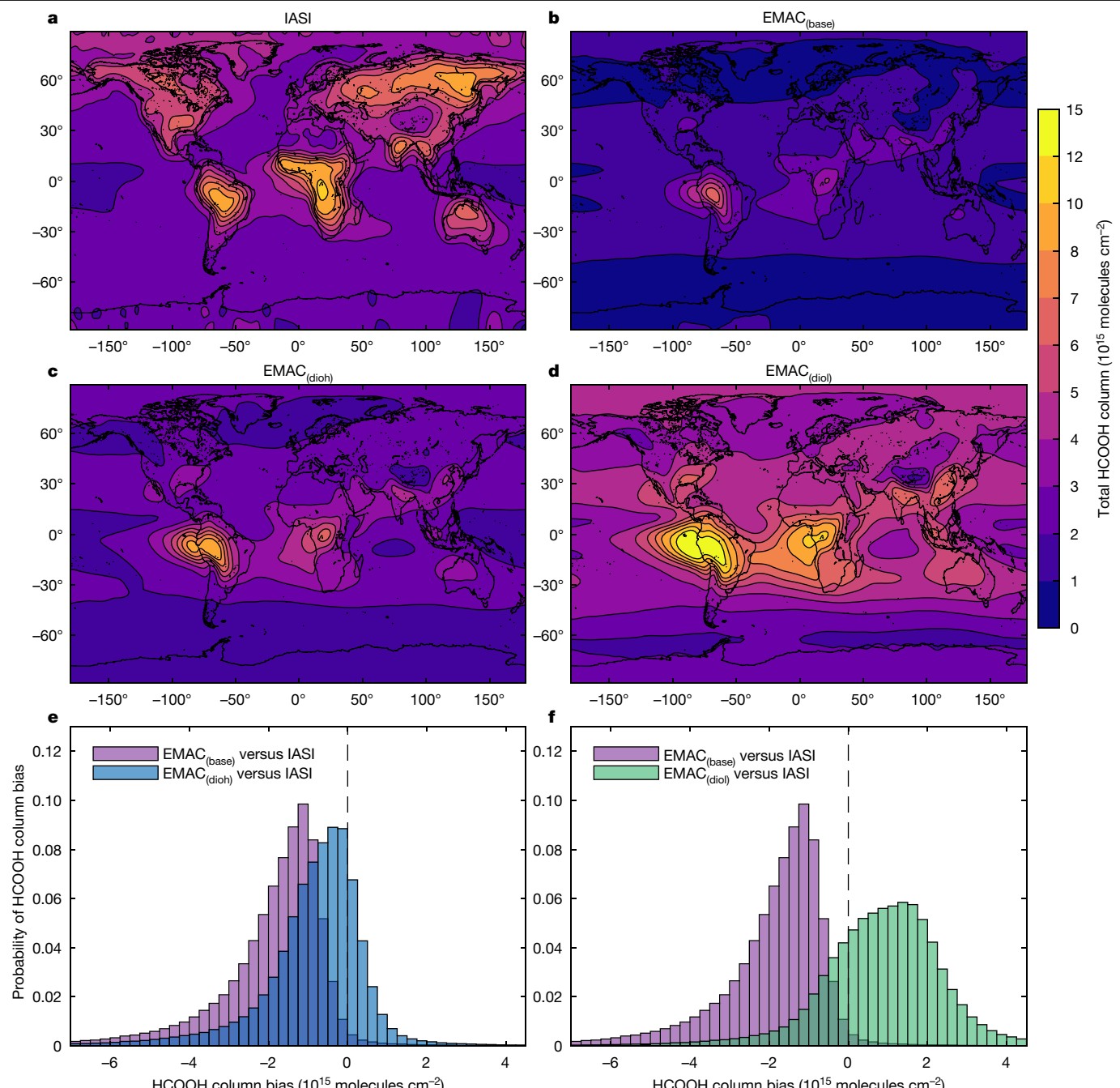

**Fig. 1 | Formic acid abundance from satellite and model. a–d,** Total formic acid (HCOOH) column (colour scale) derived from IASI satellite observations (**a**), or simulated by the base version of the model (EMAC(base); **b**) or by the model that implements the multiphase production of HCOOH (**c**, EMAC(dioh); **d**, EMAC(diol)). The HCOOH columns are means over 2010–2012. **e, f,** Probability histograms of the HCOOH column bias between EMAC simulations and satellite data. For EMAC(base) versus IASI (purple; **e, f**), the mean column bias over 2010–2012 is $-1.97 \times 10^{15}$ molecules cm$^{-2}$, the median is $-1.59 \times 10^{15}$ molecules cm$^{-2}$ and

the $1\sigma$ standard deviation is $1.64 \times 10^{15}$ molecules cm$^{-2}$. For EMAC(dioh) versus IASI (blue; **e**), the mean is $-0.88 \times 10^{15}$ molecules cm$^{-2}$, the median is $-0.66 \times 10^{15}$ molecules cm$^{-2}$ and the $1\sigma$ standard deviation is $1.62 \times 10^{15}$ molecules cm$^{-2}$. For EMAC(diol) versus IASI (green; **f**), the mean is $0.99 \times 10^{15}$ molecules cm$^{-2}$, the median is $0.97 \times 10^{15}$ molecules cm$^{-2}$ and the $1\sigma$ standard deviation is $2.16 \times 10^{15}$ molecules cm$^{-2}$. A seasonal comparison is provided in Extended Data Figs. 3, 4.

produced in this manner is efficiently oxidized by OH in the aqueous phase before outgassing. As a result, the net contribution of in-cloud HCOOH formation is small[18]. Because most methanediol is assumed to instantaneously dehydrate to formaldehyde before it volatilizes, global models do not explicitly represent methanediol and instead account for direct aqueous-phase formation of HCOOH from formaldehyde[19,20] (Fig. 2). Using experimental kinetic data[21], we calculate that under typical warm cloud conditions (260–300 K) methanediol dehydration takes

place on timescales of 100–900 s. This is longer than the timescales of cloud-droplet evaporation and aqueous-phase diffusion, which are shorter than 100 s and 0.1–0.01 s, respectively[22,23]. Moreover, methanediol transfer at the gas–liquid interface proceeds rapidly[22]. Therefore, the net flux is driven by the difference in chemical potential between the two phases. We provide evidence that methanediol reaction with OH in the gas phase quantitatively yields HCOOH under atmospheric conditions (Fig. 2). By conducting experiments with the atmospheric

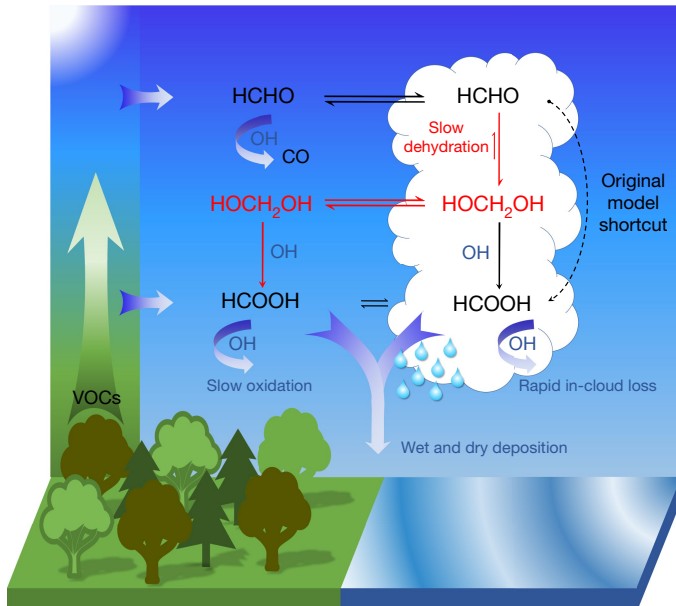

**Fig. 2 | Schematic of the multiphase production of formic acid.** The common assumption in global atmospheric chemistry models is illustrated in black: aqueous-phase methanediol ($HOCH_2OH$) is neglected and aqueous-phase formic acid (HCOOH) is assumed to form directly from formaldehyde (HCHO) on reaction with OH. The implementation of $HOCH_2OH$ multiphase equilibria is illustrated in red: the explicit representation of the slow dehydration of aqueous-phase $HOCH_2OH$, of its fast outgassing from cloud droplets and of its OH-initiated oxidation in the gas phase leads to a pervasive production of gaseous HCOOH. Under typical daytime conditions with average $[OH]_{(g)} = 1 \times 10^6$ molecules $cm^{-3}$ and $[OH]_{(aq)} = 1 \times 10^{-13}$ mol $l^{-1}$, the lifetimes of $HOCH_2OH$ against OH are about $1 \times 10^5$ s and $3 \times 10^4$ s, respectively. Under typical midday conditions with $[OH]_{(g)} = 5 \times 10^6$ molecules $cm^{-3}$, the gas-phase sink is five times stronger. Thus, gas-phase oxidation sustains the chemical gradient that drives $HOCH_2OH$ from the aqueous to the gas phase.

simulation chamber SAPHIR (Supplementary Information, section 1), we show that formaldehyde in aqueous solution is efficiently converted to gaseous methanediol immediately after injection, which quantitatively yields HCOOH on photo-oxidation (Fig. 3). This is supported by theoretical calculations (Supplementary Information, section 2). Hence, the competition between the gas- and aqueous-phase oxidation of methanediol determines the phase in which HCOOH is predominantly produced.

We implemented in EMAC the explicit kinetic model for the aqueous-phase transformations and bidirectional phase transfer of methanediol (Supplementary Information, section 3). The solubility of methanediol is not known at any temperature and estimates of it span two orders of magnitude at 298 K. We gauge the effect of this uncertainty on the results by performing the simulations EMAC(diol) and EMAC(dioh), which implement the multiphase chemistry of methanediol with Henry's law constants (solubilities) for methanediol of around $10^4$ M $atm^{-1}$ and $10^6$ M $atm^{-1}$, respectively (Methods). At the temperatures prevailing inside the clouds, the kinetic barrier strongly limits the dehydration of methanediol, allowing large amounts to be produced and then outgassed. Over regions with high levels of gas-phase formaldehyde and in the presence of clouds, large methanediol fluxes to the gas phase are predicted (Extended Data Fig. 2). Eventually, rapid gas-phase oxidation of methanediol by OH forms HCOOH, resulting in a substantial increase in the predicted HCOOH columns, by a factor of 2–4 compared to EMAC(base) (Fig. 1, Extended Data Figs. 3, 4). Because cloud droplets may potentially form everywhere and formaldehyde is ubiquitous in the troposphere (Extended Data Fig. 5), the HCOOH enhancement occurs both in high-VOC concentration regions and in remote environments.

The additional HCOOH production allows the model predictions to reach the measured HCOOH levels derived from IASI and to reduce the mean ($\pm 1\sigma$) model-to-satellite biases from $-1.97(\pm 1.64) \times 10^{15}$ molecules $cm^{-2}$ for EMAC(base) to $-0.88(\pm 1.62) \times 10^{15}$ molecules $cm^{-2}$ for EMAC(dioh) and $0.99(\pm 2.16) \times 10^{15}$ molecules $cm^{-2}$ for EMAC(diol) (Fig. 1). Similar improvements are observed with respect to the FTIR data (Extended Data Fig. 1).

Although the multiphase mechanism fills the gap between model and measurements globally, the EMAC(dioh) and EMAC(diol) simulations overpredict the HCOOH columns over tropical forests and underpredict the columns over boreal forests. We ascribe these remaining discrepancies primarily to inaccuracies in the predicted formaldehyde distributions as compared to Ozone Monitoring Instrument (OMI)/Aura measurements (Extended Data Fig. 5). Regional underestimation (overestimation) of modelled formaldehyde translates through the multiphase conversion to underprediction (overprediction) of HCOOH (Extended Data Fig. 6). For instance, underestimated biomass-burning emissions of VOCs lead to an underpredicted abundance of formaldehyde, and hence of HCOOH, such as during the 2010 Russian wildfires (Extended Data Fig. 6a–d). Conversely, the too-high model temperatures over Amazonia during the dry season induce an excess in isoprene emissions, which results in too-high formaldehyde and HCOOH levels (Extended Data Fig. 6i–l). More realistic VOC emissions, and enhanced modelling of formaldehyde and its dependence on $NO_x$, will eventually lead to further improvements in predicted HCOOH. Fast reaction of HCOOH with stabilized Criegee intermediates have recently been emphasized[24,25]. The overprediction of HCOOH over the tropical forests might be reduced if this additional sink were considered. Implementation of α-hydroperoxycarbonyls photolysis[9,26] and photo-oxidation of aromatics[27], and of a temperature-dependent solubility for methanediol, would further improve the representation of HCOOH.

We present in Table 1 a revised atmospheric budget for HCOOH, which we compare to estimates from recent studies[6–9] (the contribution of single chemical terms is provided in Extended Data Table 1). EMAC(dioh) and EMAC(diol) provide, respectively, lower and higher estimates of the extra HCOOH produced via the multiphase processing of formaldehyde. EMAC(diol) yields an increase by a factor of five of the total photochemical source predicted by EMAC(base) (190.9 Tg $yr^{-1}$ compared to 37.7 Tg $yr^{-1}$), and gas-phase oxidation of methanediol becomes the dominant contributor to atmospheric HCOOH (150.6 Tg $yr^{-1}$). Although EMAC(dioh) assumes that methanediol is 100 times more soluble (compared to EMAC(diol)), it still yields an increase by a factor of two in photochemical production (83.5 Tg $yr^{-1}$). This is in line with previous estimates of the missing HCOOH sources, which include, from source inversions, direct HCOOH emissions from vegetation or the OH-initiated oxidation of a short-lived, unidentified biogenic precursor[7]. The second largest source is VOC ozonolysis (about 31 Tg $yr^{-1}$); other sources are below 4 Tg $yr^{-1}$.

The extra HCOOH production leads to a more realistic prediction of atmospheric organic acids and substantially increases atmospheric acidity globally (Extended Data Fig. 7). Compared to EMAC(base), EMAC(dioh) and EMAC(diol) predict a decrease in the pH of clouds and rainwater in the tropics by as much as 0.2 and 0.3, respectively. The high moisture content, extended cloud cover and high temperatures that prevail in tropical and similar environments facilitate the production of HCOOH via formation and outgassing of the relevant gem-diol. Higher acidity is also predicted at North Hemisphere mid-latitudes in summertime, notably over boreal forests, consistent with previous predictions[7].

The multiphase production of HCOOH affects predictions for formaldehyde and carbon monoxide (CO). Both gases are important for tropospheric ozone and radical cycles, and are usually the target of satellite-driven inversion modelling. EMAC(dioh) and EMAC(diol) predict decreases of up to 10% and 20%, respectively, in formaldehyde columns over tropical source regions during specific months (Extended Data

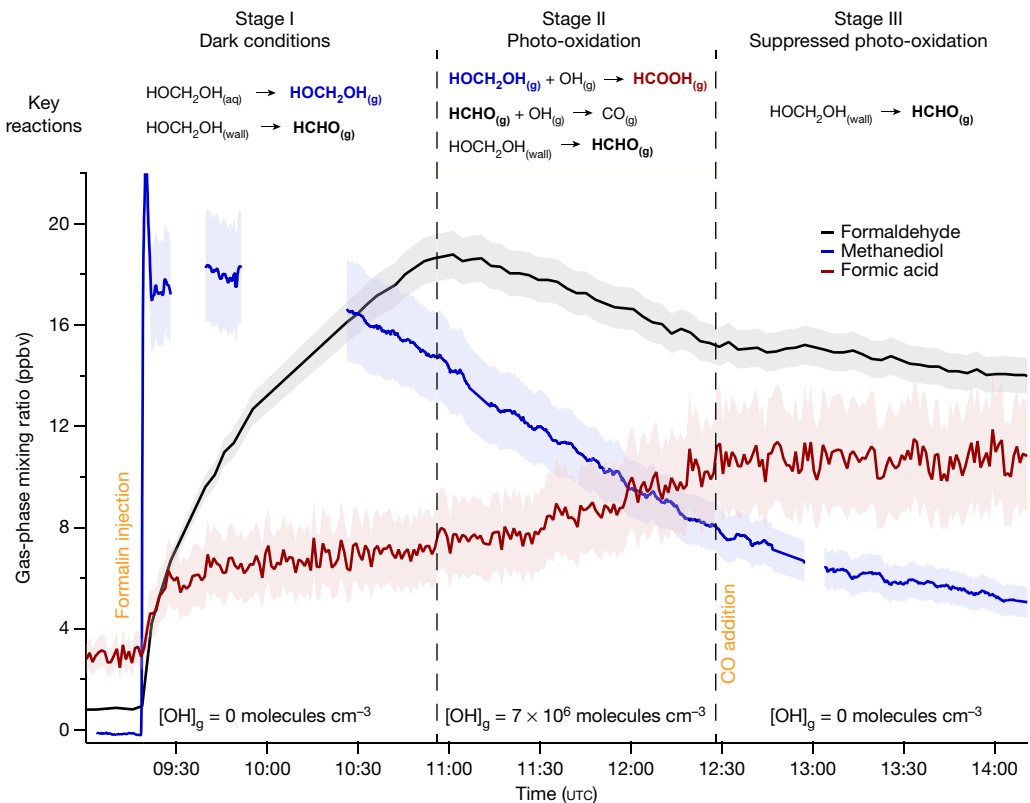

**Fig. 3 | Multiphase production of formic acid in the SAPHIR chamber.** The formaldehyde (HCHO) mixing ratio was measured (in parts per billion by volume, ppbv) by differential optical absorption spectroscopy (black), whereas the sum of HCHO and methanediol (HOCH₂OH) was measured using the Hantzsch method. The difference between the Hantzsch and differential optical absorption spectroscopy signals enables visualization of HOCH₂OH (blue). Formic acid (HCOOH) was monitored by using proton-transfer reaction time-of-flight mass spectrometry (red). The instrument uncertainties (shading) are 5% for HCHO, 12% for HOCH₂OH and 20% for HCOOH. On injection of the formalin (stabilized formaldehyde) solution into the Teflon chamber, HOCH₂OH immediately outgasses from the droplets. The chamber roof is initially closed (stage I). The gas-phase HCHO mixing ratio is initially very low, but increases to be as abundant as HOCH₂OH just before the start of the photo-oxidation when the roof is opened (stage II). The decay of the HCHO and HOCH₂OH signals is concurrent with an additional production of HCOOH. Finally, addition of carbon monoxide (CO) as an OH scavenger enabled quantification of the wall effects (stage III). Experimental details are provided in Supplementary Information, sections 1 and 4.

## Table 1 | Atmospheric budget for formic acid

| Budget terms | GEOS-Chem[6,8] | IMAGES v2[7] | MAGRITTE v1.1[9] | EMAC(base) | EMAC(dioh)–EMAC(diol) |
|---|---|---|---|---|---|
| Sources (Tg yr⁻¹) | | | | | |
| Anthropogenic | 2.3–6.3 | 4.0 | 2.2 | 2.9 | 2.9 |
| Biomass burning | 1.5 | 4.0 | 3.0 | 2.5 | 2.5 |
| Terrestrial biogenic | 2.7–4.4 | 5.6 | 5.6 | 0[a] | 0[a] |
| Photochemical | 48.6–51.0 | 88.6 | 32.9 | 37.7 | 83.5–190.9 |
| Total | 56.7–61.5 | 102.0[b] | 43.7 | 43.1 | 88.9–196.3 |
| Sinks (Tg yr⁻¹) | | | | | |
| Dry deposition | 48.8–50.6 | 33.6 | - | 10.6 | 15.7–25.5 |
| Scavenging | | 40 | | 18.6[c] | 48.6–126.4[c] |
| Photochemical | 9.5–10.6 | 28.4 | - | 13.2 | 23.6–42.7 |
| Total | 56.8–62.3 | 104 | - | 42.4 | 87.8–194.6 |
| Burden (Tg) | - | - | - | 0.55 | 1.0–1.8 |
| Lifetime (days) | 3.2 | 4.3 | - | 4.7 | 3.4–4.1 |

The table shows modelled global budget terms for formic acid (HCOOH), calculated by GEOS-Chem v8.3[6] (2004–2008 average), GEOS-Chem[8] (unknown version; 2013), IMAGES v2[7] (2009), MAGRITTE v1.1[9] (2013), EMAC v2.53.0 (EMAC(base); standard version; 2010) and EMAC v2.53.0 with the multiphase chemistry of methanediol (EMAC(dioh)–EMAC(diol); 2010). The contribution of single chemical terms is provided in Extended Data Table 1.
[a]The biogenic bidirectional fluxes from the MEGAN v2.04 model were not considered.
[b]Obtained by inverse modelling with the first IASI distribution of HCOOH[7].
[c]Net of in-cloud production and destruction and of rainout (Extended Data Table 1).

Fig. 8). We anticipate that the estimates of regional hydrocarbon emissions based on formaldehyde source inversions will be improved once the multiphase mechanism is accounted for. The reduced formaldehyde concentrations result in lower modelled CO yield from methane oxidation, notably over remote areas, where methane oxidation is the main source of atmospheric CO (Extended Data Fig. 9). Globally, the average tropospheric CO yield from methane oxidation changes from 0.91 for $EMAC_{(base)}$ to 0.88 for $EMAC_{(diol)}$ and 0.90 for $EMAC_{(dioh)}$, in agreement with isotope-enabled inversion estimates[28].

We have shown that a multiphase pathway involving aldehyde hydrates is decisive in predicting organic acid formation and atmospheric acidity. It could also be important in the presence of deliquescent aerosols and would explain the elevated HCOOH levels in cloud-free conditions[29]. Given the favourable hydration equilibrium constants for major $C_2$–$C_3$ carbonyls[30], this pathway opens up avenues for more realistic representation of other abundant organic acids, and hence of cloud-droplet nucleation and cloud evolution. We expect the multiphase processing for glyoxal and methylglyoxal to be important for explaining the observed concentrations of oxalic and pyruvic acids[4]. Understanding these multiphase processes advances our knowledge of atmospheric reactive carbon oxidation chains and of chemistry–climate interactions.

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

# Article

## Methods

### Model setup and simulations

Simulations were performed with the ECHAM5/MESSy v2.53.0 model[11] (EMAC) on the JURECA supercomputer[31]. A horizontal resolution of T63 (about 1.8° × 1.8°), with 31 vertical layers from the surface up to the lower stratosphere at 10 hPa, was applied. Chemical feedbacks are deactivated by using the quasi chemical transport mode[32]. Biomass-burning emissions are calculated with the Global Fire Assimilation System (GFAS) inventory[33]. The emission factors for organic compounds were taken from ref. [34], except the ones for aromatics, which were taken from refs. [35,36]. Anthropogenic emissions of $NO_x$ and organic compounds were taken from ACCMIP[37]. The chosen gas-phase chemical mechanism includes a state-of-the-art representation of terpene and aromatics oxidation chemistry[20]. The EMAC cloud and precipitation parameterization follows ref. [38].

In the reference model simulation ($EMAC_{(base)}$), HCOOH production proceeds through the ozonolysis of alkenes with terminal double bonds (simple alkenes and degradation products of isoprene and monoterpenes), alkyne oxidation, reaction of formaldehyde with the peroxy radical, oxidation of enols, and formation from vinyl alcohol[39]. Nonetheless, we exclude the OH-initiated oxidation of isoprene and monoterpenes, the corresponding mechanisms of which are still speculative[6,8,40,41], as well as the reaction of methyl peroxy radical with OH, which was shown not to yield HCOOH[42]. A detailed description of the relevant chemical kinetics, budget terms and deposition parameters for each model simulation is provided in Supplementary Information, section 3a.

Two simulations with the explicit multiphase model for methanediol, $EMAC_{(dioh)}$ and $EMAC_{(diol)}$, are described in detail in Supplementary Information, section 3b. The simulations differ only by the value of the Henry's law constant (solubility) of methanediol, for which no experimental measurements are available. Values of about $10^4$ M atm$^{-1}$ and $10^6$ M atm$^{-1}$ are used for $EMAC_{(diol)}$ and $EMAC_{(dioh)}$, respectively. These are possible values of the Henry's law constant for methanediol, given the spread of estimates at 298 K by semi-empirical methods and the expected temperature dependence. However, higher values (around $10^7$ M atm$^{-1}$) cannot be excluded at typical temperatures of warm clouds (Supplementary Information, section 3b.iii).

For the comparison with IASI and OMI observations (Fig. 1, Extended Data Figs. 3–6), the HCOOH and formaldehyde volume mixing ratio profiles simulated by EMAC are sampled along the Sun-synchronous satellite Metop-A and Aura orbits, respectively, at the time and location of the IASI and OMI measurements, using the SORBIT submodel[11]. The sampled volume mixing ratios are then daily averaged and computed in HCOOH and formaldehyde columns.

Model sources of uncertainties, including the formation of a HCOOH·H$_2$O complex with water vapour[43], are discussed in Supplementary Information, section 5.

### IASI column observations

IASI[44] is a nadir-viewing Fourier transform spectrometer launched on board the Metop-A, -B and -C platforms in October 2006, September 2012 and November 2018, respectively. IASI measures in the thermal infrared, between 645 cm$^{-1}$ and 2,760 cm$^{-1}$. It records radiance from the Earth's surface and the atmosphere, with an apodized spectral resolution of 0.5 cm$^{-1}$, spectrally sampled at 0.25 cm$^{-1}$. In the spectral range in which the HCOOH $\nu_6$ Q branch absorbs (about 1,105 cm$^{-1}$), IASI has a radiometric noise of around 0.15 K for a reference blackbody at 280 K. IASI provides near global coverage twice per day, with observations at around 09:30 am and 09:30 pm, local time. Here, the HCOOH columns are derived from IASI/Metop-A (covering 2010–2012). Only the morning satellite overpasses are used, because such observations have a higher measurement sensitivity[13]. For comparison with EMAC simulations, the 2010–2012 IASI data are daily averaged on the model spatial grid. On average, 17 satellite measurements per day (more than 18,000 over

2010–2012) are used per 1.8° × 1.8° model grid box at the Equator. This number increases with latitude and with the higher spatial sampling of IASI, owing to the satellite polar orbits.

Version 3 of the artificial neural network for IASI (ANNI) was applied to retrieve HCOOH abundances from the IASI measurements (see refs. [13,45] for a comprehensive description of the retrieval algorithm and the HCOOH product). The ANNI framework was specifically designed to provide a robust and unbiased retrieval of weakly absorbing trace gases such as HCOOH. The retrieval relies on a neural network to convert weak spectral signatures to a total column, accounting for the state of the surface and atmosphere at the time and place of the overpass of IASI. The vertical sensitivity of IASI to HCOOH peaks between 1 km and 6 km, gradually decreasing outside that range[46]. However, by assuming that HCOOH is distributed vertically according to a certain profile, the neural network is able to provide an estimate of the total column of HCOOH. Because the ANNI retrievals do not rely on a priori information, no averaging kernels are produced and the retrieved columns are meant to be used at face value for carrying out unbiased comparisons with model data (see ref. [13] and references therein for the rationale). Data filtering prevents retrieval over cloudy scenes and post-filtering discards scenes for which the sensitivity to HCOOH is too low for a meaningful retrieval.

The HCOOH product comes with its own pixel-dependent estimate of random uncertainties, calculated by propagating the uncertainties of each input variable of the neural network[13]. For a typical non-background HCOOH abundance (($(0.3–2.0) \times 10^{15}$ molecules cm$^{-2}$), the relative uncertainty on an individual retrieved column ranges from 10% to 50%, with the highest uncertainties found for the low columns. This uncertainty increases for lower-background columns as the weaker HCOOH concentrations approach the IASI detection threshold. However, these random uncertainties become negligible for the column averages presented here, because of the total number of measurements per grid cell. With respect to systematic uncertainties, the main term is related to the assumption of a fixed HCOOH vertical profile. It is not possible to quantify this uncertainty on an individual-pixel basis, but it was estimated to not exceed 20% on average[13]. A comparison with independent HCOOH columns from ground-based FTIR measurements at various latitudes and environments confirmed the absence of any large systematic biases of the IASI data[45]. Although biases of around 20% cannot be excluded, in the context of this work, the accuracy of the IASI product is sufficient to demonstrate the initial model underprediction ($EMAC_{(base)}$) of the HCOOH columns and the large improvements from the multiphase mechanism.

### Theoretical predictions

Quantum chemical calculations were performed at various levels of theory, up to CCSD(T)/CBS(DTQ)//IRCMax(CCSD(T)//M06-2X/aug-cc-pVQZ), and combined with E,J-µVTST multi-conformer microvariational transition-state calculations to obtain rate coefficients for the gas-phase high-pressure-limit rate coefficients (Supplementary Information, section 2).

### Data availability

The EMAC model data are publicly accessible at https://doi.org/10.5281/zenodo.4315292, https://doi.org/10.5281/zenodo.4315276 and https://doi.org/10.5281/zenodo.4314730. The IASI measurements may be found at https://doi.org/10.5281/zenodo.4314367. The OMI measurements are openly distributed via the Quality Assurance for Essential Climate Variables repository (https://doi.org/10.18758/71021031). The FTIR observations are publicly accessible at https://doi.org/10.5281/zenodo.4321348 and https://doi.org/10.5445/IR/1000127831. Data from the experiments are available on the Eurochamp database (https://doi.org/10.25326/Q00C-MY65, https://doi.org/10.25326/KHYY-FP10, https://doi.org/10.25326/BC4N-TY93 and https://doi.org/10.25326/DAS4-7Q54).

The raw quantum chemical data are provided in Supplementary Information, section 10. Source data are provided with this paper.

## Code availability

The Modular Earth Submodel System (MESSy) is continuously being developed and applied by a consortium of institutions. The usage of MESSy and access to the source code is licensed to all affiliates of institutions that are members of the MESSy Consortium. Institutions can become a member of the MESSy Consortium by signing the MESSy Memorandum of Understanding (more information at http://www.messy-interface.org). The modifications presented here were implemented on MESSy v2.53.0. The source code used to produce the results is archived at the Jülich Supercomputing Centre and can be made available to members of the MESSy community on request.

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

**Acknowledgements** We acknowledge the computing time granted by the JARA-HPC Vergabegremium and VSR commission on the supercomputer JURECA at Forschungszentrum Jülich. B.F. was partly supported by a Marie Curie COFUND postdoctoral fellow grant co-funded by the University of Liège and the European Union (FP7-PEOPLE, project ID 600405). L.C. and E.M. are research associates with the F.R.S.–FNRS (Brussels). The IASI mission is a joint mission of EUMETSAT and the Centre National d'Etudes Spatiales (CNES, France). This work received funding from the Initiative and Networking Fund of the Helmholtz Association through the projects 'Advanced Earth System Modelling Capacity (ESM)' and 'Pilot Lab Exascale Earth System Modelling (PL-ExaESM)'. The content of this paper is the sole responsibility of the author(s) and it does not represent the opinion of the Helmholtz Association, and the Helmholtz Association is not responsible for any use that might be made of the information contained. Most IASI activities were supported by the Belgian State Federal Office for Scientific, Technical and Cultural Affairs (Prodex arrangement IASI.FLOW). OMI HCHO developments at BIRA were supported by the EU FP7 QA4ECV project, in close cooperation with KNMI, University of Bremen, MPIC-Mainz and WUR (http://www.qa4ecv.eu/ ecv/hcho-p). The FTIR monitoring program of ULiège at Jungfraujoch benefited from the involvement of C. Servais and O. Flock. It was primarily funded by the F.R.S.–FNRS (grant no. J.0147.18), the Fédération Wallonie-Bruxelles and the GAW-CH program of MeteoSwiss. We acknowledge the International Foundation High Altitude Research Stations Jungfraujoch and Gornergrat (HFSJG, Bern) for supporting the facilities needed to perform the FTIR observations at Jungfraujoch. Eureka FTIR measurements were made at the Polar Environment Atmospheric Research Laboratory (PEARL) by the Canadian Network for the Detection of Atmospheric Change (CANDAC), led by J. R. Drummond, and in part by the Canadian Arctic ACE/OSIRIS Validation Campaigns, led by K. A. Walker; support was provided by AIF/NSRIT, CFI, CFCAS, CSA, ECCC, GOC-IPY, NSERC, NSTP, OIT, PCSP and ORF. Toronto FTIR measurements were made at the University of Toronto Atmospheric Observatory, supported by CSA, NSERC, ECCC and UofT. The National Center for Atmospheric Research is sponsored by the National Science Foundation. The NCAR FTS observation programmes at Thule (GR) and Boulder (CO) are supported under contract by NASA. The Thule work is also supported by the NSF Office of Polar Programs (OPP). We thank the Danish Meteorological Institute for support at the Thule site. The Wollongong solar remote sensing program has been supported through a series of Australian Research Council grants, most recently DP160101598. Funding via Helmholtz ATMO programme has enabled sustained NDACC FTIR activities at Kiruna and Izana since the late 1990s. The Izana NDACC FTIR observations strongly rely on the support (facilities and operational activities) of the Izana Atmospheric Research Centre of the Spanish Weather Service (AEMET), with lead contributions of O. E. García. Researchers from KIT thank U. Raffalski and P. Voelger (Swedish Institute of Space Physics; IRF) for continuing support of the NDACC FTIR site Kiruna. We thank Université de La Réunion and CNRS (LACy-UMR8105 and UMS3365) for financial support of the FTIR instrumentation at St-Denis, and C. Hermans, F. Scolas (BIRA-IASB) and J.-M. Metzger (Université de La Réunion) for FTIR maintenance. The IASI HCOOH data and FTIR HCOOH data at St-Denis are currently obtained through the BRAIN-be project OCTAVE, financed by the Belgian Science Policy Office (BELSPO).

**Author contributions** B.F. and D.T. initiated and coordinated the study, designed and performed the EMAC simulations, performed the data analyses, prepared the figures and wrote the manuscript. D.T., R.S. and S.R. developed and implemented the multiphase mechanism for formaldehyde. S.G. prepared the initial setup of the model, provided the tools to obtain the formic acid budget and CO yield from methane oxidation. T.E. and A.K. implemented the revised dry deposition scheme. C. Cho, H.-P.D., H.F., G.G., T.H., A.N., D.R., R.T., D.T. and L.V. conducted, analysed and described the SAPHIR experiments; L.V. performed and described the theoretical calculations for the gas-phase methanediol oxidation. B.F., L.C., M.P., C. Clerbaux and P.-F.C. performed the IASI retrievals and provided the IASI dataset. I.D.S. and M.V.R. provided the OMI dataset and expertise on OMI data usage. E.M., T.B., M.D.M., D.W.T.G., J.W.H., F.H., N.J., E.L., I.O., C.P.-W., M.S., K.S. and C.V. contributed to instrument operation, performed the retrievals and/or provided the FTIR datasets. A.K.-S., A.W., A.P., L.C., P.-F.C. and E.M. contributed to discussions of the results and preparation of the manuscript.

**Funding** Open access funding provided by Forschungszentrum Jülich GmbH.

**Competing interests** The authors declare no competing interests.

**Additional information**
**Correspondence and requests for materials** should be addressed to B.F. or D.T.

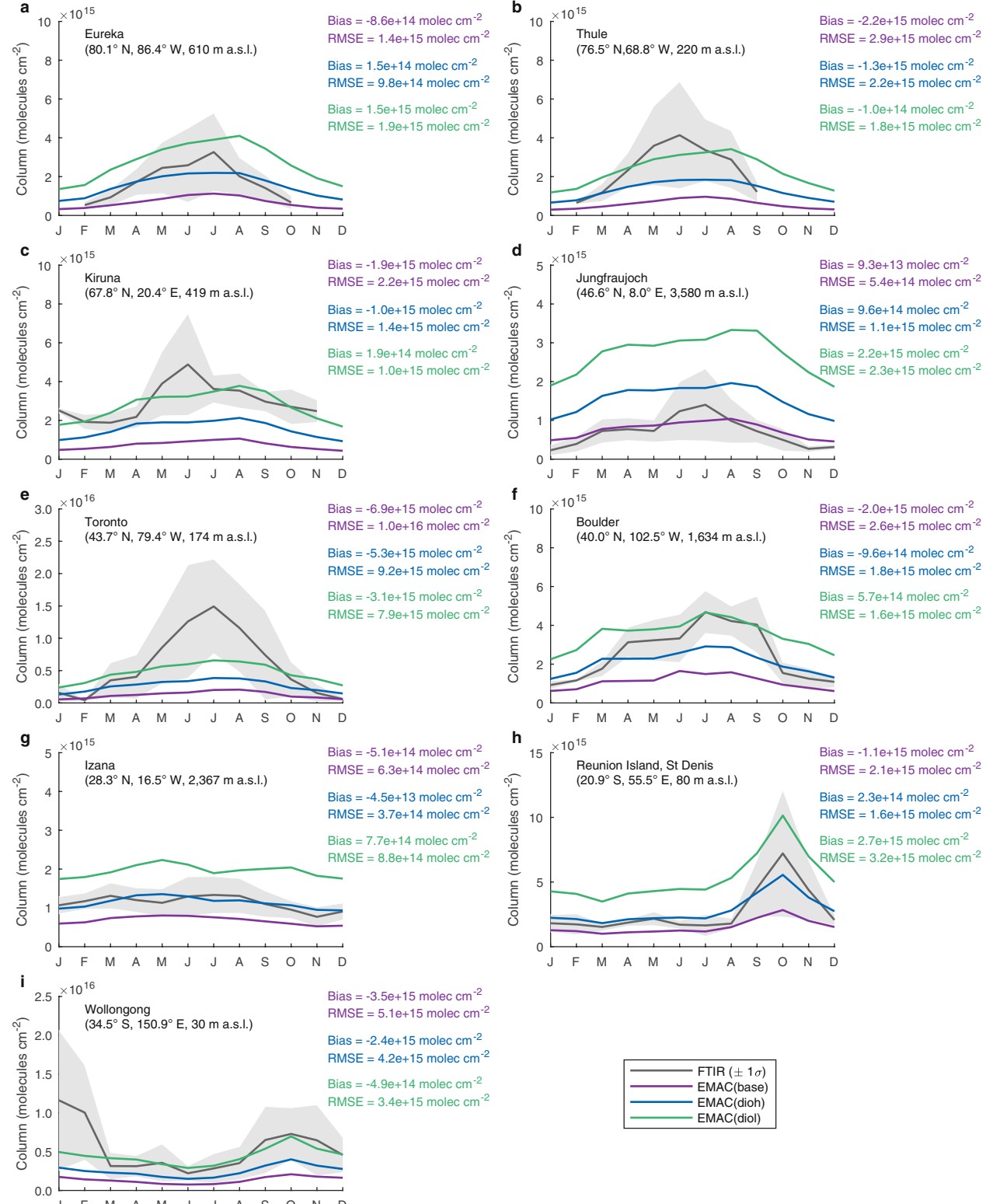

**Extended Data Fig. 1 | Seasonal cycle of formic acid from FTIR and model.**
**a–i**, Formic acid (HCOOH) monthly average columns at nine different FTIR stations, displayed on a 1-year time base, from the 2010–2012 ground-based FTIR observations and EMAC simulations. The grey shaded areas correspond to the 1σ standard deviation of the individual FTIR measurements around the monthly average. The mean column bias and root-mean-squared error (RMSE) were calculated between the daily mean FTIR and EMAC data, over the days with FTIR measurements available. The vertical sensitivity of the FTIR retrievals was accounted for by applying averaging kernels (except at Wollongong, where no averaging kernels were produced). Details on the ground-based FTIR retrievals are provided in Supplementary Information, section 6. m a.s.l., metres above sea level.

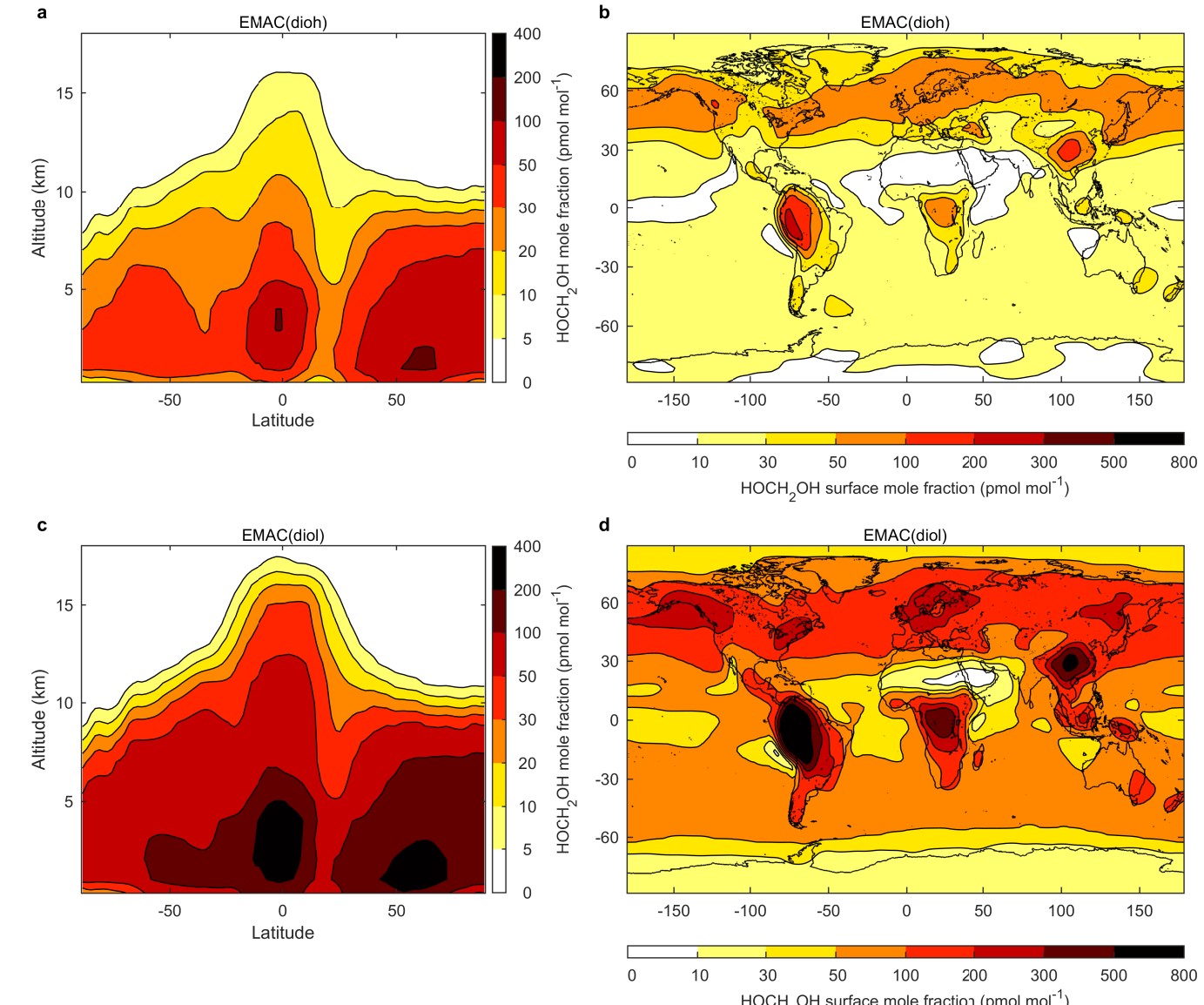

**Extended Data Fig. 2 | Global methanediol distribution simulated by EMAC. a–d**, Zonal mean (**a**, **c**) and surface (**b**, **d**) methanediol (HOCH₂OH) mole fraction simulated by EMAC(dioh) (**a**, **b**) and EMAC(diol) (**c**, **d**) over 2010–2012. The EMAC(dioh) and EMAC(diol) simulations implement the multiphase chemistry of methanediol. On reaction with OH in the gas phase, methanediol yields formic acid (HCOOH).

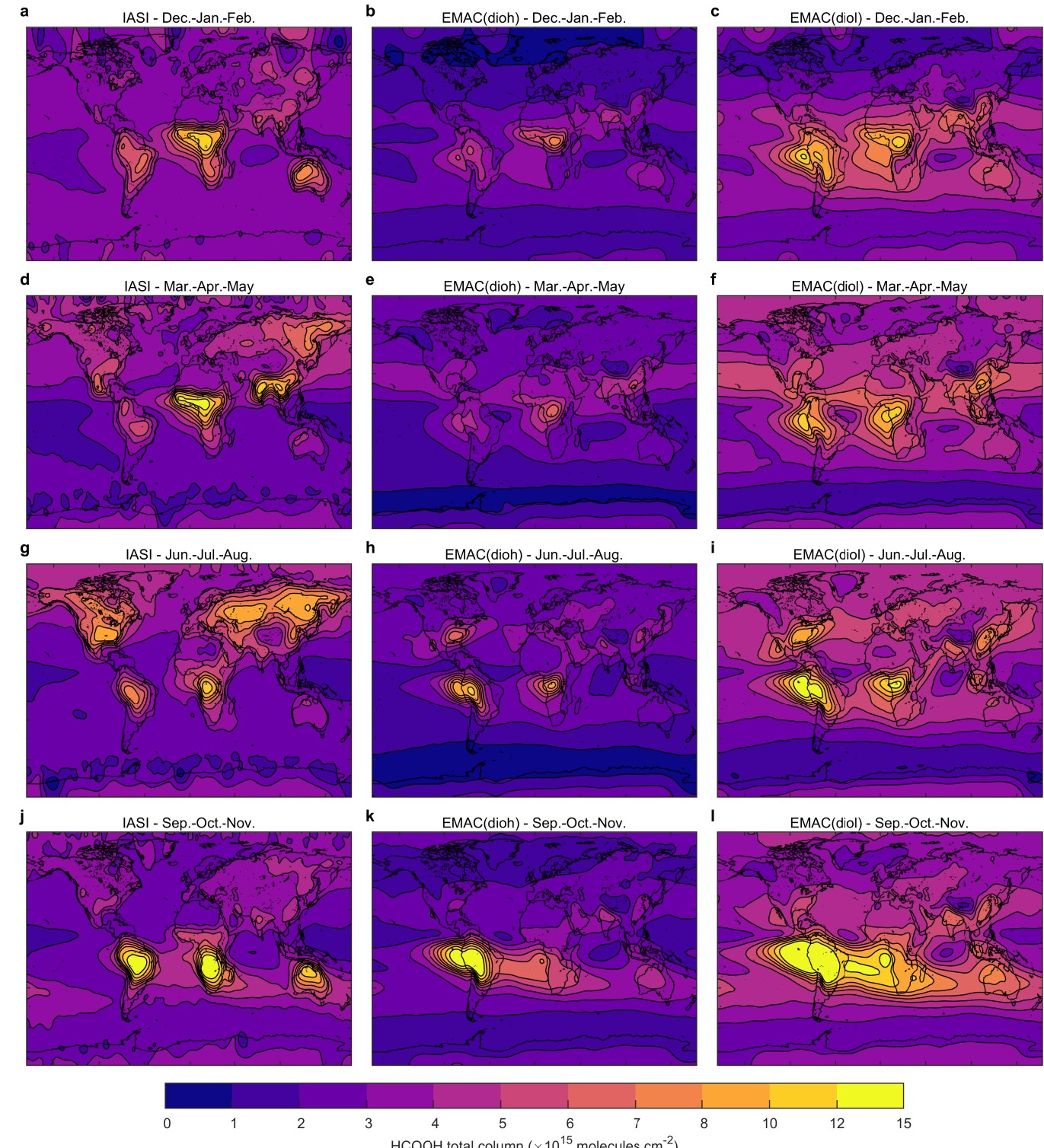

**Extended Data Fig. 3 | Global formic acid abundance from satellite and model. a–l,** Formic acid (HCOOH) column derived from IASI satellite observations (**a, d, g, j**), and simulated by the EMAC model that implements the additional production of HCOOH via the multiphase chemistry of methanediol (**b, e, h, k**, EMAC$_{(dioh)}$; **c, f, i, l**, EMAC$_{(diol)}$). Model data were sampled at the time and location of the satellite measurements. The total columns are seasonal averages over December–February (**a–c**), March–May (**d–f**), June–August (**g–i**) and September–November (**j–l**) 2010–2012. Statistics on the EMAC-to-IASI HCOOH column biases are presented in Extended Data Fig. 4.

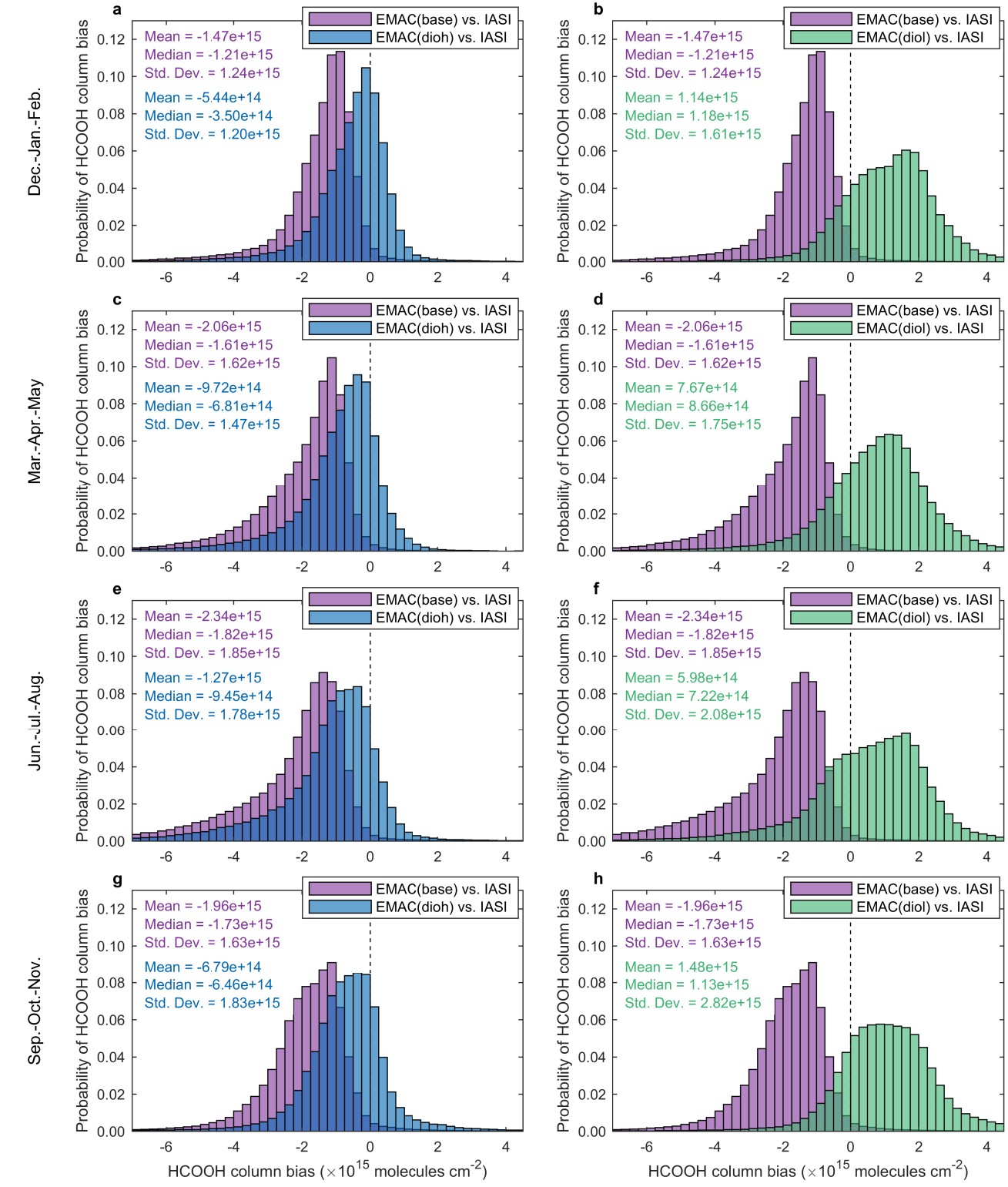

**Extended Data Fig. 4 | Formic acid column biases between model and satellite. a**–**h**, Probability histograms of the seasonal formic acid (HCOOH) column bias between EMAC simulations and IASI satellite data over December–February (**a**, **b**), March–May (**c**, **d**), June–August (**e**, **f**) and September– November (**g**, **h**) 2010–2012. The statistics correspond to the mean, median and 1σ standard deviation of the column biases calculated between the EMAC and IASI columns for each season. The associated global HCOOH column distributions are displayed in Extended Data Fig. 3.

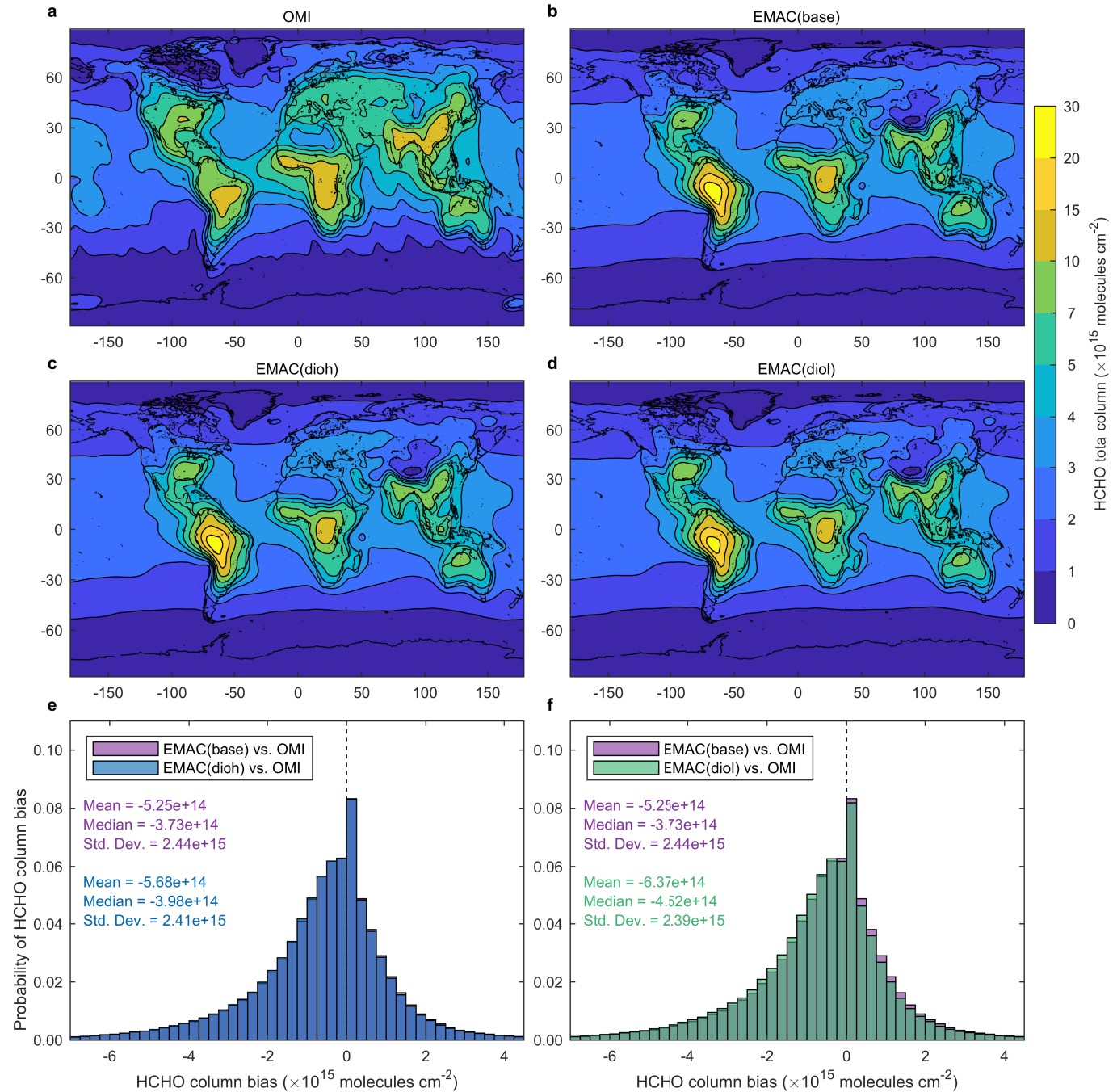

**Extended Data Fig. 5 | Global formaldehyde abundance from satellite and model. a–d**, Formaldehyde (HCHO) column derived from OMI/Aura satellite observations (**a**), or simulated by EMAC(base) (**b**), EMAC(dioh) (**c**) or EMAC(diol) (**d**). Model data were sampled at the time and location of the satellite measurements, and the OMI averaging kernels were applied to the model profiles to account for the vertical sensitivity and resolution of OMI. The HCHO columns are means over 2010–2012. **e**, **f**, Probability histograms of the HCHO column bias between EMAC simulations and satellite data. The statistics correspond to the mean, median and $1\sigma$ standard deviation of the column biases calculated over 2010–2012. Details on the OMI HCHO retrievals and the comparison with model data are provided in Supplementary Information, section 7.

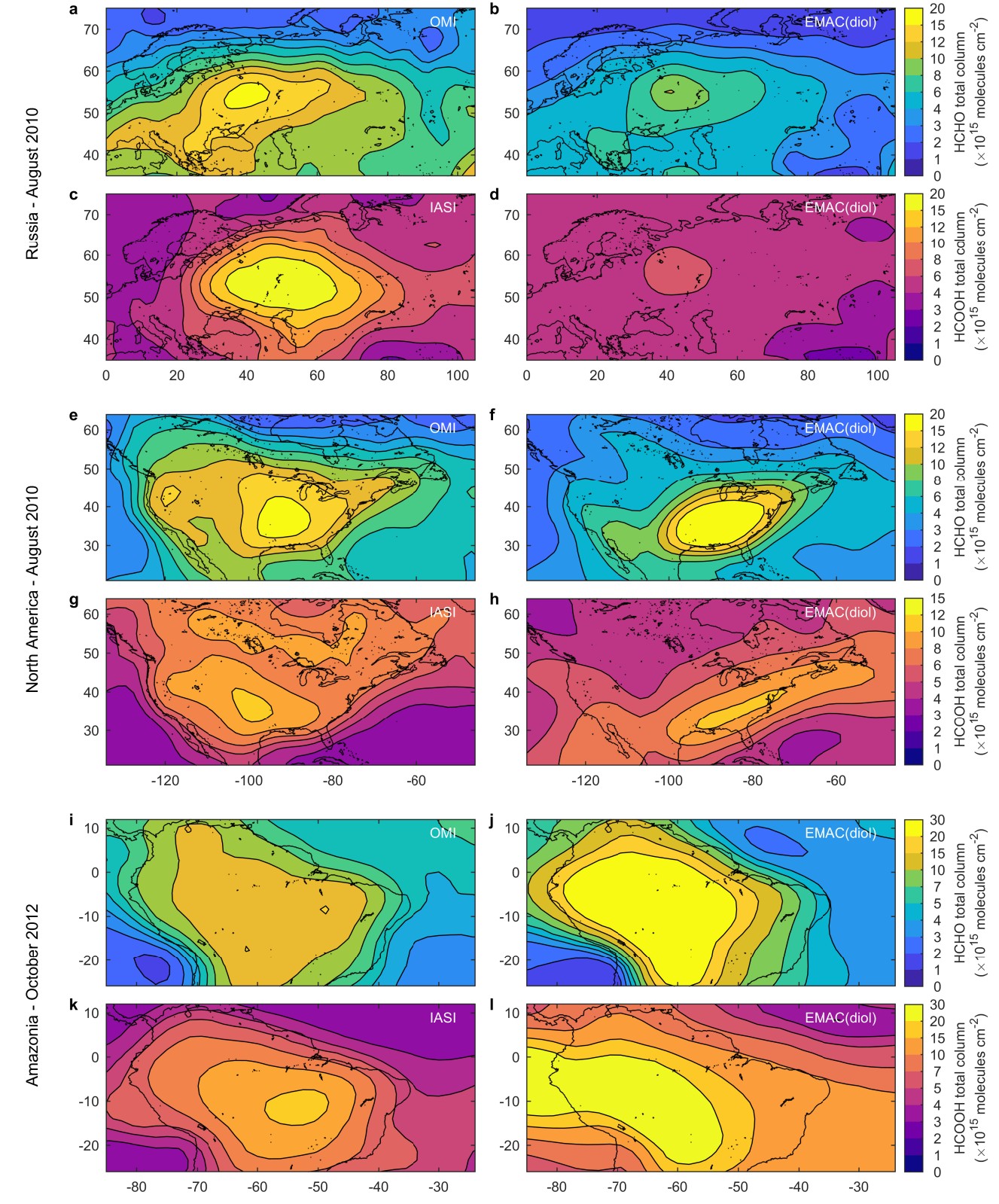

**Extended Data Fig. 6 | Effect of modelled formaldehyde biases on formic acid prediction. a–l,** Monthly average formaldehyde (HCHO; **a**, **b**, **e**, **f**, **i**, **j**) and formic acid (HCOOH; **c**, **d**, **g**, **h**, **k**, **l**) columns from IASI and OMI satellite measurements (**a**, **c**, **e**, **g**, **i**, **k**), respectively, and from the EMAC$_{(diol)}$ simulation (**b**, **d**, **f**, **h**, **j**, **l**), over Russia in August 2010 (**a**–**d**), North America in August 2012 (**e**–**h**) and Amazonia in October 2010 (**i**–**l**). HCHO and HCOOH model data were sampled at the time and location of the OMI and IASI satellite measurements, respectively. The OMI averaging kernels were applied to the model profiles to account for the vertical sensitivity and resolution of OMI (IASI averaging kernels are not available). The same comparison, but for EMAC$_{(dioh)}$, is provided in Supplementary Fig. 7 (Supplementary Information, section 8).

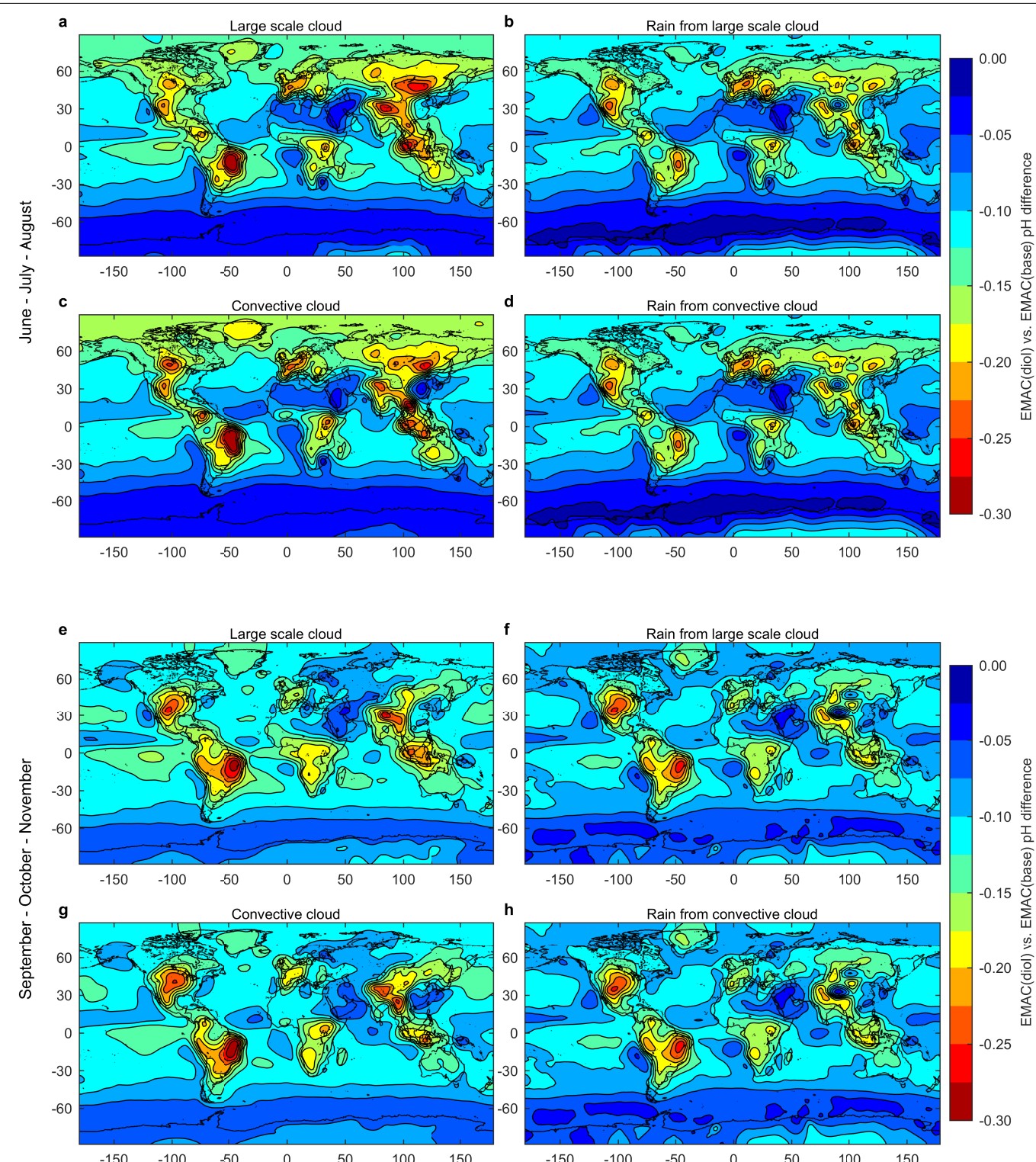

**Extended Data Fig. 7 | Effect of cloud processing on cloud and rainwater acidity. a**–**h**, pH difference of the large-scale clouds (**a**, **e**) and associated rain (**b**, **f**), and of the convective clouds (**c**, **g**) and associated rain (**d**, **h**), between the EMAC$_{(diol)}$ and EMAC$_{(base)}$ simulations. The pH differences are seasonal averages over June–August (**a**–**d**) and September–November (**e**–**h**) 2010–2012. The pH decrease is due to the additional production of formic acid (HCOOH) via the multiphase chemistry of methanediol implemented in EMAC$_{(diol)}$. The effect on cloud and rain pH of the EMAC$_{(dioh)}$ simulation is displayed in Supplementary Fig. 8 (Supplementary Information, section 8).

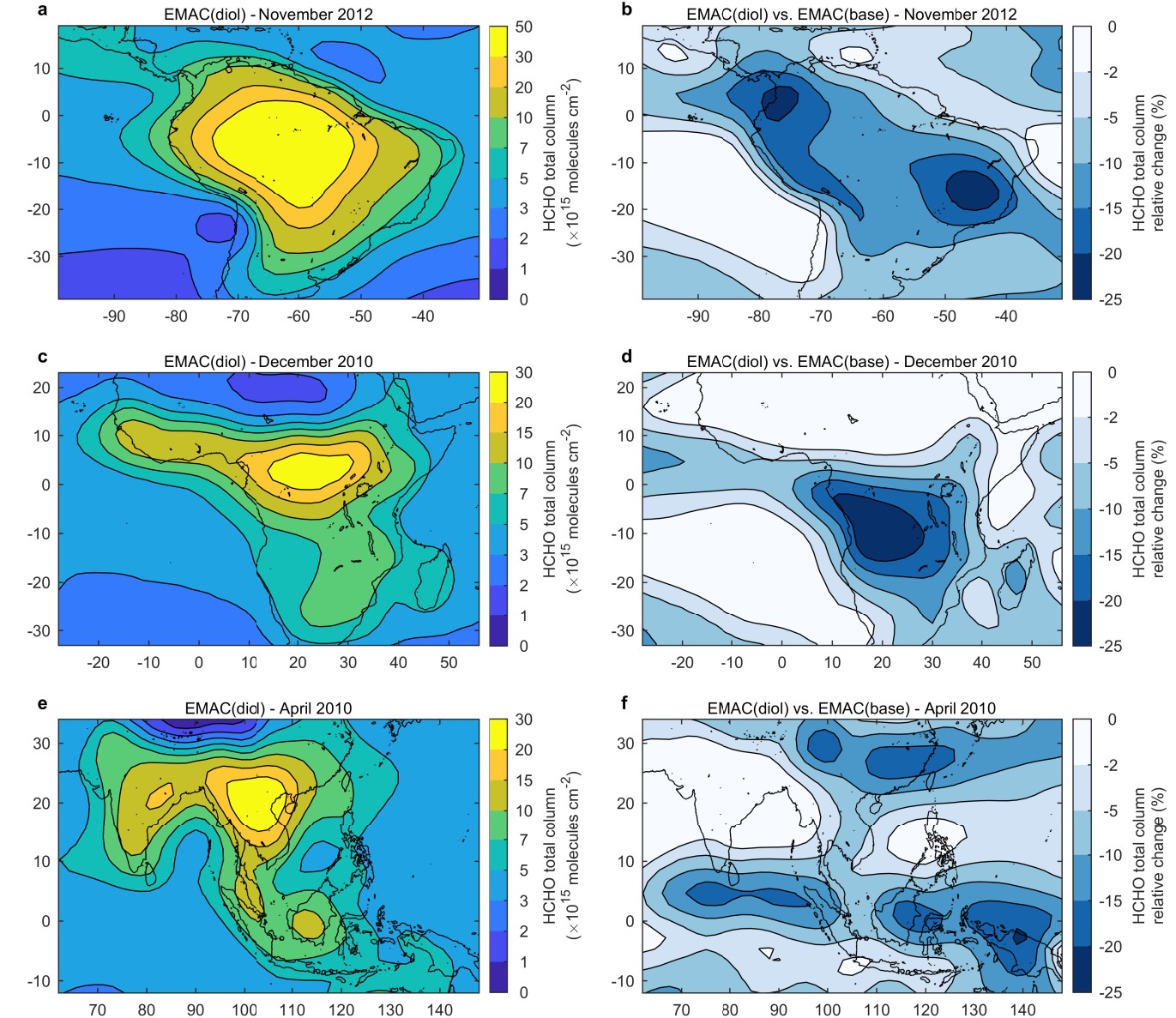

**Extended Data Fig. 8 | Effect of cloud processing on formaldehyde modelling.** Monthly average formaldehyde (HCHO) total column simulated by EMAC(diol) (**a**, **c**, **e**), and relative difference in HCHO total column between EMAC(diol) and EMAC(base) (**b**, **d**, **f**), over Amazonia in November 2012 (**a**, **b**), central Africa in December 2010 (**c**, **d**) and southeast Asia in April 2010 (**e**, **f**). The effect on HCHO modelling of the EMAC(dioh) simulation is presented in Supplementary Fig. 9 (Supplementary Information, section 8).

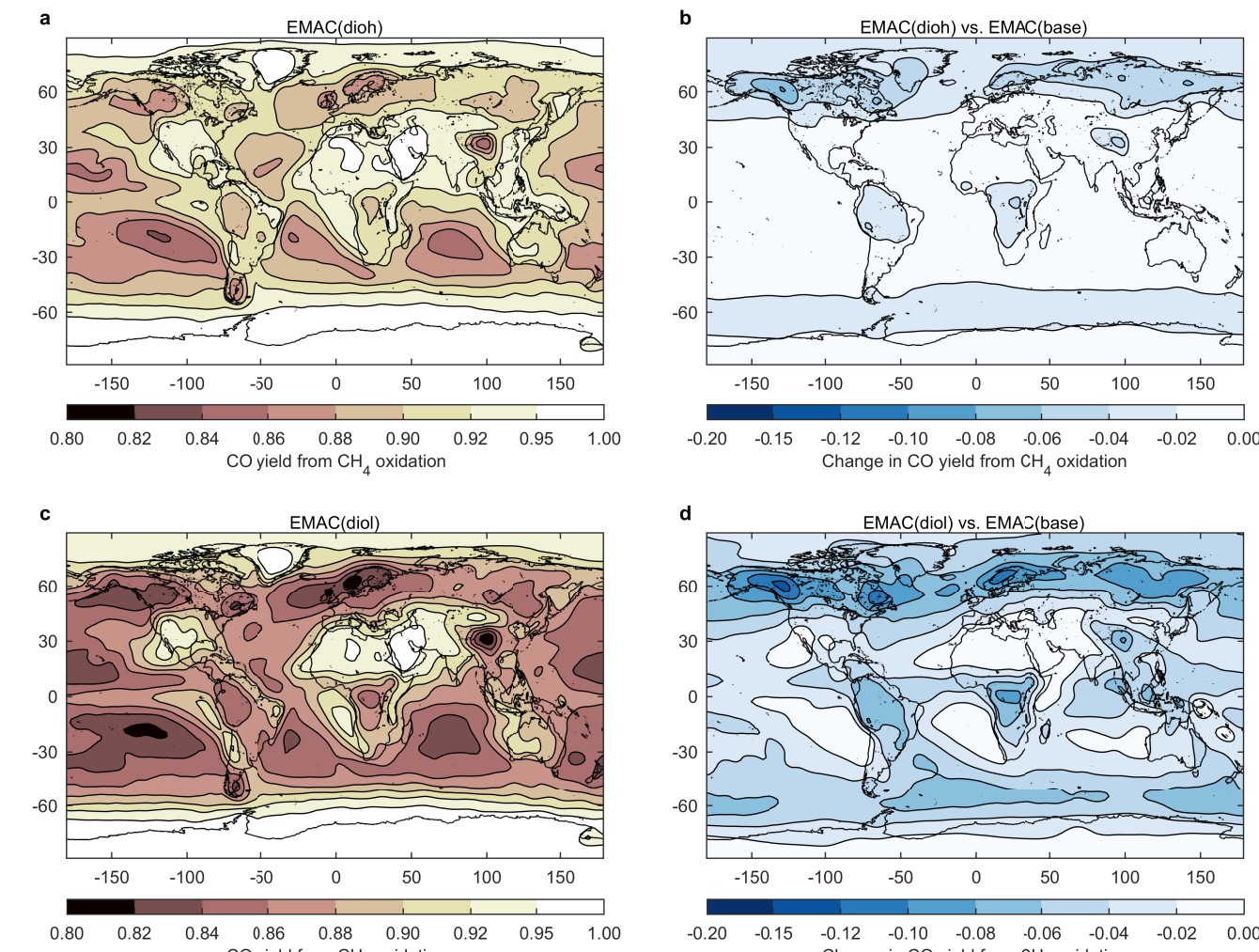

**Extended Data Fig. 9 | Effect of cloud processing on modelled carbon monoxide. a–d**, Yield of CO originating from methane (CH₄) oxidation modelled by EMAC$_{(dioh)}$ (**a**) and EMAC$_{(diol)}$ (**c**), and change in CO yield from CH₄ oxidation between EMAC$_{(dioh)}$ (**b**) or EMAC$_{(diol)}$ (**d**) and EMAC$_{(base)}$. The data presented are annual averages over 2010–2012.

**Extended Data Table 1 | Atmospheric chemical budget of formic acid calculated by EMAC**

| Budget terms (Tg yr$^{-1}$) | EMAC$_{(base)}$ | EMAC$_{(dioh)}$ | EMAC$_{(diol)}$ |
|---|---|---|---|
| **Sources** | | | |
| $HOCH_2OH_{(g)} + OH_{(g)} \rightarrow$ | 0 | 44.9 | 150.6 |
| $HOCH_2O_{2(g)} + HO_{2(g)} \rightarrow$ | 1.9 | 1.9 | 1.9 |
| $Alkenes_{(g)} + O_{3(g)} \rightarrow$ | 28.7 | 29.9 | 31.9 |
| $HC{\equiv}CH_{(g)} + OH_{(g)} \rightarrow$ | 3.2 | 3.2 | 3.2 |
| $H_2C{=}CHOH_{(g)} + OH_{(g)} \rightarrow$ | 4.0 | 3.6 | 3.3 |
| Total | 37.7 | 83.5 | 190.9 |
| **Sinks** | | | |
| $HCOOH_{(g)} + OH_{(g)} \rightarrow$ | 13.2 | 23.6 | 42.7 |
| $HOCH_2OH_{(aq)} + OH_{(aq)} \rightarrow$ $HCOOH_{(aq)} + HO_{2(aq)} \rightarrow$ $HCOO^-_{(aq)} + OH_{(aq)} \rightarrow$ $HCOOH_{(aq)} + OH_{(aq)} \rightarrow$ | 18.6* | 48.6* | 126.4* |
| Total | 31.8 | 62.2 | 169.1 |

Modelled global terms of formic acid (HCOOH) calculated by EMAC$_{(base)}$, EMAC$_{(dioh)}$ and EMAC$_{(diol)}$ for the year 2010. EMAC$_{(dioh)}$ and EMAC$_{(diol)}$ account for a higher and a lower solubility of methanediol, respectively (Supplementary Information, section 3).
*Net loss of HCOOH for the reactions list including a small contribution by rainout.