## [Peer Review File · Nature]

Manuscript Title: Ubiquitous atmospheric production of organic acids mediated by warm clouds

Reviewer Comments & Author Rebuttals**Reviewer Reports on the Initial Version:****Ref #1**

Franco et al argue for a new chemical source of formic acid (HCOOH) that helps to reconcile the gap between theoretical and observed atmospheric measurements. The new source is from the oxidation of methanediol (HOCH₂OH), formaldehyde (HCHO) monohydrate, which exists in cloud water.

Their argument is based on the competing timescales between HOCH₂OH dehydrating before it volatilizes to return HCHO, and cloud drop evaporation and aqueous phase diffusion. Current calculations assume that HOCH₂OH dehydrates instantaneously to return HCHO. However, under typical warm cloud conditions, evaporation and aqueous phase diffusion in combination occur a factor 10 times quicker than dehydration. This new formation pathway leads to large additional source of HCOOH.

The authors then proceed to explore the consequences of this new source on global tropospheric chemistry. They show that this new source of HCOOH not only helps to reconcile theoretical model and remote sensing measurements HCOOH but also discrepancies in HCHO and carbon monoxide.

It is very interesting hypothesis that is testable with new observations that will undoubtedly be collected in the light of this paper. I think the paper would benefit from a few more details and a more rigorous treatment of statistics, which is sadly lacking.

The warm cloud is the vehicle responsible for this additional HCOOH. Given the temperature dependence of the aqueous and gas-phase reactions it would be useful for this reader to understand the temperature range the authors had in mind. The authors did not discuss the role of pH, but the from the Winkelman reaction rates appear to be valid over a wide range of pH values. It would be useful if the author commented on this.

The authors have chosen to evaluate their HCOOH findings using remote sensing data. These data provide large-evidence for the model bias. The satellite data comparison shown is a mean over three years, with no statistics reported. How uncertain are HCOOH retrievals? Did the author take the mean between 0930 and 2130? There is some discussion but the error sources as discussed appear to be large. Do I understand correctly that in the absence of scene-dependent averaging kernels the authors have compared IASI HCOOH column with the model averaged between 1 and 6 km? Given the large model minus observation mismatch I doubt whether it will make difference whether they compare over different altitudes but the authors should make that statement.

Over the spatial resolution of the model (approximately 1,800 km over the tropics) do the author think they over- or under- estimate temperature of clouds? This is related to my question about the temperature dependence of the reactions. Typically how many IASI observations are used per grid box? What is the spin-up time for the model? Does comparing EMAC(diols) make sense in 2010?

As a reader I am curious about how this comparison changes during the year, e.g. wet (cloudy) and dry (smoky) seasons over the tropics where there are large enhancements observed by IASI. Certainly, the authors should provide some statistical evidence of the better fit to data even if it is obvious that the model bias has been greatly reduced.

I am also curious, why they have chosen to focus exclusively on remote sensing column data. There are many recent European and NASA aircraft experiments that include HCOOH, HCHO, CO, etc that may provide additional constraints on the hypothesis being proposed.

The authors make a comment about inferring isoprene emissions from HCHO columns using an inverse model. They may well be right with a Bayesian approach. If their hypothesis is correct the additional HCOOH source would be larger during the wet season when there is extensive cloud cover and HCHO cannot be observed using UV wavelengths. I suggest more statistical evidence is produced on a seasonal timescale is produced before they can make a statement about casting doubt on these studies. Even with a 30% reduction in model columns (as they state) due to the additional HCOOH source the model still has a large positive bias compared to OMI (Figure 6 in extended data).

Similarly, the authors argue that the additional HCOOH sources helps to improve the agreement with MOPITT and in situ CO measurements. I am unconvinced with the analysis as currently reported. In Figure 7 in the extended data, the dramatic changes in colour are somewhat misleading. A change in yellow to red could simply be a change of 10%. A clear statistical report would go a long way to supporting their case. The comparison with in situ data is even less convincing. Without accompanying statistics I cannot comment further. However, based on the plots provided I suspect reporting meaningful changes in ground-based CO is probably a bit of a stretch.

Ref #2

This is a relatively straightforward paper about the atmospheric chemistry of formic acid. Several recent studies have highlighted that formic acid is more abundant in the atmosphere than currently known sources can account for. The paper uses some previously published kinetic data to argue that methanediol, formed from formaldehyde hydrolysis in cloud droplets, can partition to the gas phase, where it is oxidized by OH to form formic acid. The mechanism is included in a global model and the resulting formic acid columns are compared with those from the IASI satellite instrument and from ground-based FTIR column measurements. Including the mechanism gives a better description of the observations, particularly in regions with strong isoprene emissions, and also lowers the atmospheric burdens of carbon monoxide and formaldehyde. The Authors argue that this brings the model results for these species also in better agreement with the CO and formaldehyde observations. Finally, the Authors discuss other atmospheric implications of these findings, including the acidity of cloud and rain water, and the atmospheric burdens of sulfur dioxide and hydrogen peroxide.

I find the paper interesting and important, but have to admit that I am not completely overwhelmed by the evidence. Figure 1 shows that the new mechanism can account for higher formic acid over regions with high biogenic emissions, but the IASI data also show high columns at northern mid latitudes and over the boreal forests in North America and Russia. There is virtually no discussion of this in the paper. In addition, given the laboratory facilities available to the Authors, I had expected a new experimental confirmation of the proposed mechanism, or an atmospheric measurement of methanediol. While the Authors discuss the lack of methanediol measurements in the SI, there was no effort to characterize how existing instruments respond to this compound. Having said that, the proposed mechanism does provide a potential and new explanation for a long-standing problem in atmospheric chemistry, and the publication of this paper would spur a lot of new activity to evaluate these findings.

Detailed comments:

The manuscript does not devote much attention to the role of nitrogen oxides. One issue with formic acid has been that several of the known formation mechanisms are most efficient in low

NO_x conditions, whereas efficient formic acid formation has also been observed at high NO_x. I encourage the Authors to consider this aspect of their work in more detail.

In several instances, I believe that the paper could do more to explore alternative explanations of its findings. For example, biomass burning provides the largest direct emission source of formic acid and these emissions are difficult to describe accurately in models. There is barely any mention of this source and its uncertain emissions in the paper. Could the high formic acid columns observed by IASI over the boreal forests in the summer be due to wildfires?

Likewise, the paper is quick to conclude that lower formation sources of CO and formaldehyde in the model are more consistent with the measurements, but there are many other reasons that could account for these differences, including uncertain emissions of CO and of the biogenic precursors of CO and formaldehyde, oxidant concentrations, simplified chemical mechanisms in global models, etc.

It is interesting that the mechanism can also account for acetic acid formation from acetaldehyde. In this case, however, acetaldehyde measurements have been higher than global models can account for, and this issue would presumably only get worse if the acetaldehyde to acetic acid pathway were included in a global model. Some discussion of this would be good to add.

Figure 1: Are the formic acid columns shown in panels A-C actually comparing the same thing? As discussed in the SI, the satellite instrument is more sensitive to the mid troposphere than to the surface. A brief discussion in the main body of the paper of the averaging kernel of the instrument, and to what extent this has, or has not, been taken into account in the comparison with the model output would help to better understand and appreciate this graph.

Ref #3

Review of "Ubiquitous atmospheric production of organic acids mediated by warm clouds" by Franco, et al. for publication in Nature:

This is a highly intriguing manuscript suggesting that the hydration of formaldehyde within cloud-water to form methylene glycol (HOCH₂OH, methanediol) and subsequent evaporation and gas-phase oxidation thereof by OH radical forming formic acid to be an important, or rather, the dominant source (more than 4 times greater than all other known sources combined) of formic acid to the atmosphere. The authors suggest this chemistry changes our understanding of the pH in clouds and rain water, particularly in the tropical regions, through significant increases in acidity due to higher levels of formic acid produced by this mechanism (conversion of formaldehyde to formic acid). The authors also suggest similar chemistry involving larger aldehydes may be important for organic aerosol formation and growth.

Understanding the levels of organic acids, particularly the abundant formic and acetic acids in the atmosphere has been a long-standing problem in the field of atmospheric chemistry. Here the authors of the present work present a hypothesis which can more than explain the observed abundance of formic acid in the atmosphere. While the hypothesis presented here is indeed interesting, logical and plausible, and worthy of continued work, and even publishable as a hypothesis-type paper within a more specialized journal, it is not clear that this manuscript, as it currently stands, meets the criteria for publication in a high profile type of journal such as Nature, due primarily to the large uncertainties inherent in the methods used to validate the hypothesis – uncertainties, which in my opinion, preclude quantitative assessment of the importance of the suggested chemistry for the atmosphere. These uncertainties are discussed further below.

Henry's Law Constant constant of HOCH₂OH: The authors use a value of $1.0 \times 10^4 \text{ M atm}^{-1}$ which is obtained from the computer software HENRYWIN provided by the US EPA using a bond or group specific structure additivity relationship (SAR) to calculate the water solubility of specific

molecules. While this software produces solubility results in reasonable agreement with experimental results for some species, it also produces quite unreliable results for others. Therefore, one should assign a high uncertainty toward results from this SAR, when no experimental values exist (i.e., it is not inconceivable that the Henry's Law constant for this species could be of order 50 times greater than the number used within these simulations). For instance, the bond method from HENRYWIN calculates ethylene glycol solubility to be about 80 times smaller than recommended by the JPL Kinetics Panel [Burkholder, et al. 2015]. A similar error for methylene glycol solubility would result in the same increase in the ratio of methylene glycol in the condensed to that in the gas phase, and presumably have an important impact on the amount of formic acid produced from the subsequent gas-phase HOCH₂OH oxidation. The authors clearly need to discuss the uncertainty in this quantity and the sensitivity of their results toward this.

Depositional loss of HOCH₂OH: The authors do not discuss the importance of dry (and wet) deposition for methylene glycol. Instead only the 'rapid gas-phase oxidation [by OH]' is discussed. Though, the assumed OH rate constant, $1.3 \times 10^{-11} \text{ cm}^3 \text{ molec}^{-1} \text{ s}^{-1}$, yields an average OH lifetime of methylene glycol of ~21 hours for $[\text{OH}] = 1 \times 10^6 \text{ molec cm}^{-3}$. This is comparable to dry deposition timescale within the terrestrial boundary layer. Inclusion of this process may be important.

Gas phase dehydration of HOCH₂OH(g): While it is well-established that the barrier to uncatalyzed dehydration of methylene glycol in the gas phase is large (~43 kcal/mol), precluding this reaction from being important under atmospherically relevant temperatures, calculations have indicated that this process can be catalyzed by a number of other species [e.g., Kumar and Francisco, 2015, and references therein]. Given the scarcity of atmospheric work on this species it is surprising the authors did not include a discussion this paper and references therein in this manuscript. Investigation of the potential importance of catalyzed dehydration of methylene glycol is necessary to determine the robustness of the author's assumption that OH + methylene glycol(g) will be the dominant fate of this species within the atmosphere.

OH + HOCH₂OH rate constant uncertainty: Needed to address both of the previous comments is the potential uncertainty in the assumed OH + methylene glycol rate constant. This rate coefficient has not been measured and is simply calculated here using an SAR which has no validation for gem-diol type functional groups. This is likely uncertain by a factor of 2-3.

An experimental study on the gas phase OH oxidation chemistry (kinetics and products) of methylene glycol would go a long way toward reducing the uncertainties discussed above, and in my opinion, is a needed prerequisite to publish this work in a journal such as Nature.

The authors use comparisons of formic acid column abundance from chemical transport model simulations and remote sensing observations (ground and satellite spectroscopic measurements) to help support their hypothesis. As stated above, it is clear that simulations using traditional chemistry substantially under-predicts formic acid in the atmosphere. Simulations using the proposed mechanism, however, substantially over-predict formic acid in many regions (particularly in the tropics) relative to the observations. Perhaps uncertainties in assumptions may play a role here? In addition, the spatial correspondence between the model and satellite column measurements is not particularly convincing, especially in the summertime over 40N-80N region (figure 1). What is the reason behind this? It may be helpful to perform a 'tagged' formic acid analysis where formic acid from various sources is tracked separately, enabling the parsing of discrepancies as a function of source.

With regard to comparisons of model simulations using the proposed mechanism of formic acid with NDACC station observations: In the remote regions the model often predicts substantially higher formic acid than is observed by the ground-based FTIR measurements. A robust understanding of these differences in the remote areas may be fruitful, as the chemistry should be

much simpler. Also, the seasonal shape is often somewhat different between model and observations. Are these things understood? Why is the model not averaged over the same time period as the observations? This can and should be done. Why is Wollongong averaging kernel treated differently?

In addition, the authors use comparisons of model simulations of HCHO and CO against satellite observations to support the hypothesis of the paper. However, these are fairly weak constraints, on the proposed mechanism, as modeled HCHO abundance is highly sensitive to VOC emissions, particular oxidation mechanisms, and chemical environment in which the oxidation occurs (i.e., NO levels), all of which are uncertain in the model realm. The changes in CO between simulations is small, and not well-distinguishing as compared with the data.

Finally, it has been observed that model/observation discrepancies between formic and acetic acids are often both similar in magnitude and correlated [Paulot, et al, 2009]. Is this coincidence? The hypothesis put forward here only helps to resolve the formic acid discrepancy. How does one make sense of this?

References:

Burkholder, J. B. et al. Chemical Kinetics and Photochemical Data for Use in Atmospheric Studies, Evaluation No. 18, JPL Publication 15-10, Jet Propulsion Laboratory, Pasadena, 2015
<http://jpldataeval.jpl.nasa.gov>.

Kumar, M., and Francisco, J. S. The role of catalysis in alkanediol decomposition: Implications for general detection of alkanediols and their formation in the atmosphere. *J. Phys. Chem. A*, 119, 9821-9833, 2015.

Paulot, F., et al. Importance of secondary sources in the atmospheric budgets of formic and acetic acids. *Atmos. Chem. Phys.*, 11, 1989-2013, 2011.

Author Rebuttals to Initial Comments:

We thank all the referees for considering our manuscript and for their thoughtful, extensive review of our study.

First we would like to apologize for the long time before the submission of a revised version of the study. The referees' comments really helped us to clarify various aspects of our research work and we put much effort into addressing them in the most comprehensive way. In that framework, we considered the referees' suggestions to bring more evidence as to the chemical mechanism that we are proposing. In particular, we performed chamber experiments, theoretical calculations, additional model simulations and comparisons with a new satellite dataset, which explains the delay of our answer. This additional work provided us with new results that confirm our previous findings and strengthen our main conclusions. We believe that we have now addressed satisfactorily the referees' concerns, and we are happy to present a revised version of the manuscript. Please find in this document the point-by-point reply (in blue) to the referees' comments.

Compared to the initial version of the study, the main changes are:

- Experiments were performed at the atmospheric simulation chamber SAPHIR providing direct evidence of methanediol formation and of the subsequent production of formic acid, consistent with the proposed multiphase mechanism. These experiments also allowed us to constrain the reactivity of methanediol with OH, yielding formic acid. Furthermore, we corroborated these findings with theoretical calculations. These results are summarized in the main text and detailed in the Supplementary Information.
- We have performed a new set of global model simulations implementing the key physicochemical parameters obtained from the experiments. We demonstrate that the model assuming a much higher solubility for methanediol still predicts, via the multiphase pathway, an important production of formic acid that is bigger than all the previously-known chemical sources combined. The simulations with the high and low solubility of methanediol gauge the range of additional formic acid that can be produced via this new source. We have revisited the global budget of formic acid accordingly. Compared with the simulations initially performed, the predicted additional production of formic acid is of the same order of magnitude, although slightly reduced.
- To evaluate the model performance, we are now using a new IASI satellite-derived formic acid product. This product was developed outside of the context of this paper (Franco et al. 2018, 2020), and as part of a larger class of VOC products from IASI. Compared to the old version, the retrieval is both more accurate and precise. One of the strengths of the product is that it provides estimates of formic acid that are not affected by a priori columns. Comparison with ground-based FTIR data confirmed the absence of any large systematic biases. We estimate remaining biases to be an order of magnitude lower than the initial model under-prediction of the formic acid columns and therefore sufficiently accurate to demonstrate the large improvements brought with the new mechanism. In the Methods, we provide a summary of the retrieval and the new IASI data.
- We have improved the model-to-observations comparisons by adding a better statistical treatment of the data. In a synthetic way, this demonstrates that the additional formic acid production via the multiphase mechanism allows the model predictions to fill the gap with independent measurements. Although there is an overall agreement with the measurements, regional discrepancies remain, such as over boreal forests. However, most of these can be explained by inaccuracies in the modelled distribution of formaldehyde, which is at the core of the new formic acid production pathway.

We would like to thank the referees for considering the revised version of this study.

Sincerely,

Bruno FRANCO and Domenico TARABORRELLI

On behalf of all co-authors.

Referee #1 (Remarks to the Author):

Franco et al argue for a new chemical source of formic acid (HCOOH) that helps to reconcile the gap between theoretical and observed atmospheric measurements. The new source is from the oxidation of methanediol (HOCH₂OH), formaldehyde (HCHO) monohydrate, which exists in cloud water.

Their argument is based on the competing timescales between HOCH₂OH dehydrating before it volatilizes to return HCHO, and cloud drop evaporation and aqueous phase diffusion. Current calculations assume that HOCH₂OH dehydrates instantaneously to return HCHO. However, under typical warm cloud conditions, evaporation and aqueous phase diffusion in combination occur a factor 10 times quicker than dehydration. This new formation pathway leads to large additional source of HCOOH.

The authors then proceed to explore the consequences of this new source on global tropospheric chemistry. They show that this new source of HCOOH not only helps to reconcile theoretical model and remote sensing measurements HCOOH but also discrepancies in HCHO and carbon monoxide.

It is very interesting hypothesis that is testable with new observations that will undoubtedly be collected in the light of this paper. I think the paper would benefit from a few more details and a more rigorous treatment of statistics, which is sadly lacking.

Thank you very much for your assessment of the paper and the detailed comments. We agree with the main comment and have now included statistics to better characterize the model-to-measurements comparisons. This helps to better appraise the improvements brought by the multiphase chemistry to the prediction of HCOOH by the global model. In particular, we refer to Fig. 1 (in the main text) and Extended Data Fig. 4 for more statistics on the model-to-IASI HCOOH comparison, to Extended Data Fig. 1 for the model-to-FTIR HCOOH comparison, and to Extended Data Fig. 5 for the model-to-OMI HCHO comparison.

The warm cloud is the vehicle responsible for this additional HCOOH. Given the temperature dependence of the aqueous and gas-phase reactions it would be useful for this reader to understand the temperature range the authors had in mind. The authors did not discuss the role of pH, but the from the Winkelman reaction rates appear to be valid over a wide range of pH values. It would be useful if the author commented on this.

Thank you for these comments. We fully agree and have added the following text to the Supplementary Information (Sect. 3.b.ii) describing the EMAC(*diol*) simulation.

“The temperature range where most of the cloud-mediated formic acid production occurs is 260-300K. However, the kinetic data used here for the hydration/dehydration of methanediol are derived from experiments in the 293-333K range. Furthermore, it is known that acid catalysis of methanediol dehydration is important for solution pH < 3.5 (Funderburck et al., 1978; Boyce and Hoffmann, 1984). This is outside the typical pH range of cloud droplets which is entirely covered (pH 5-7) by the kinetic data we use in this study (Winkelman et al., 2002). Specifically, Winkelman et al. (2000) showed that the dehydration constant of methanediol did not change significantly (<13%) for the pH 6-7.5 range.”

The authors have chosen to evaluate their HCOOH findings using remote sensing data. These data provide large-evidence for the model bias. The satellite data comparison shown is a mean over three years, with no statistics reported. How uncertain are HCOOH retrievals? Did the author take the mean between 0930 and 2130? There is some discussion but the error sources as discussed appear to be large. Do I understand correctly that in the absence of scene-dependent averaging kernels the authors have compared IASI HCOOH column with the model averaged between 1 and 6 km? Given the large model minus observation mismatch I doubt whether it will make difference whether they compare over different altitudes but the authors should make that statement.

As mentioned in the introduction to our replies, we now exploit a new satellite product for HCOOH, obtained with a neural network-based retrieval approach. Specifically, the retrieval method consists in quantifying, for each IASI observation, the amplitude of the HCOOH signal, and converting it directly into HCOOH total column using an artificial feedforward neural network (NN) (Franco et al., 2018). This NN is fed by other IASI observed parameters to account explicitly for the state of the surface and atmosphere, and assumes a vertical profile of HCOOH to convert the spectral signal into total column. The NN does not rely directly on a priori information (conversely to optimal estimation-like retrieval methods), hence no averaging kernels are produced. Indeed, though the IASI sensitivity to HCOOH is maximal in the mid-troposphere and decreases outside that range (Pommier et al., 2016), the NN is informed of the state of the atmosphere (via the temperature and humidity profiles from IASI) and therefore can deduce how much each atmospheric layer contributes to the signal as observed by IASI. The NN has been trained to account for this non-homogenous vertical signal, and to produce a HCOOH total column that can be compared directly with other total column data e.g., from global models. The disadvantage of this approach is that a fixed vertical profile shape has to be assumed. Fortunately, the associated uncertainty that is introduced is modest, as explained below.

Regarding the retrieval uncertainties, the estimated uncertainty on each input variable is propagated through the NN to yield an uncertainty on each single-pixel retrieved column (Franco et al., 2018). For a typical non-background HCOOH abundance ($0.3\text{-}2.0 \times 10^{15}$ molecules cm^{-2}), the relative uncertainty on an individual retrieved column ranges from 10 to 50%, with the highest uncertainties found for the low columns. This uncertainty increases for lower background columns. However, these uncertainties mostly cancel out by averaging multiple IASI measurements on the EMAC spatial grid. For instance, around 17 satellite measurements are averaged per day on one single $\sim 1.8^\circ \times 1.8^\circ$ model grid box at the Equator, which corresponds to >18,000 measurements over 2010-2012. This number increases

with the latitude and the higher spatial sampling of IASI due to its polar orbit. The main source of uncertainty is actually the shape of the HCOOH vertical profile that is assumed in the retrieval. Franco et al. (2018) evaluated this uncertainty to be of the order of 20% by comparing two estimates of the true HCOOH columns obtained in turn by two NNs assuming a completely different shape of the profile. A comparison with independent HCOOH columns from ground-based FTIR measurements at various latitudes and environments did not reveal any large systematic biases of the IASI data (Franco et al., 2020). We estimate remaining biases to be an order of magnitude lower than the initial model underprediction ($EMAC_{(base)}$) of the HCOOH columns and therefore sufficiently accurate to demonstrate the large improvements brought with the new mechanism. In the Methods, we provide a summary of the retrieval and of the main information on the new IASI data.

As for which data was used: here, only the a.m. measurements (~09:30 local time) corresponding to the morning satellite overpasses are exploited since they benefit from a higher sensitivity. As for the EMAC model data, those have been sampled by the SORBIT submodel (Jöckel et al., 2010) along the Metop-A satellite orbits and at the overpassing time of the satellite, which allows us to perform a more realistic comparison with the IASI columns. This information has been added in the Methods of the manuscript.

Please note that the HCOOH total columns derived from the ground-based FTIR observations are retrieved using an optimal estimation-based method. Therefore, in that case averaging kernels are produced and applied subsequently to smooth the model profiles for the evaluation of the model simulations (SI Sect. 6). The random and systematic uncertainties affecting the retrieved total columns are in the range of 11-13 and 15-18%, respectively. As with the model-to-satellite comparison, these uncertainties are much smaller than the initial model underprediction of the FTIR HCOOH columns (ED Fig. 1), and hence do not affect any of the conclusions of the study.

Over the spatial resolution of the model (approximately 1,800 km over the tropics) do the author think they over- or under- estimate temperature of clouds? This is related to my question about the temperature dependence of the reactions.

We have run our final simulations at T63 resolution, which corresponds to about 200 km grid box size in the tropics. The chosen model setup is a Quasi-Chemical Transport Mode (QCTM) for which the feedbacks between atmospheric composition and weather are switched off. We make use of meteorological nudging to ERA-Interim reanalysis data. The default nudging does not foresee the “imposition” of the global mean temperature which leads to a significant temperature bias of the EMAC model (Jöckel et al., 2016). However, for the lower troposphere where the warm clouds in question are, the EMAC model shows a relatively small negative bias of around 1K on an annual mean basis (see Jöckel et al., 2016, Fig. 12, panel “RC1SD-base-09-ERA- Interim”). Nevertheless, as we mention in the main manuscript, the time scales of cloud droplet evaporation and aqueous-phase diffusion of tracers are significantly shorter than the dehydration time scale for methanediol. Therefore, we expect a secondary role of the temperature bias in the prediction of outgassing efficiency of methanediol.

Typically how many IASI observations are used per grid box?

As mentioned above, around 17 IASI measurements per day are used on average per model grid box at the Equator, which represents >18,000 measurements over 2010-2012. Because of the satellite polar orbits, the IASI spatial sampling (and the number of observations per model grid box) increases with latitude. Averaging in space and time multiple satellite measurements per model grid box reduces significantly the measurement uncertainties. We have added this information to the Methods in the manuscript.

What is the spin-up time for the model? Does comparing EMAC(diol) make sense in 2010?

The whole year 2009 is spin-up time in the model simulations and data from 2010 on are analysed. One-year spin-up for the tropospheric configuration of the model is generally accepted as safe in our modelling community. For the EMAC_(diol) simulation, we can see that the HCOOH burden strongly increases during the whole of 2009 and shows an ample seasonal cycle throughout the 2010-2012 time period.

As a reader I am curious about how this comparison changes during the year, e.g. wet (cloudy) and dry (smoky) seasons over the tropics where there are large enhancements observed by IASI. Certainly, the authors should provide some statistical evidence of the better fit to data even if it is obvious that the model bias has been greatly reduced.

We agree that a statistical treatment really helps to appraise in a synthetic way the improvement brought by the new multiphase pathway. As for the model-to-satellite comparisons of HCOOH on an annual (Fig. 1) and seasonal basis (ED Fig. 3), we now provide histograms showing the probability distribution of the model-to-IASI column biases (in Fig. 1 and ED Fig. 4). Such histograms allow us to appraise in a synthetic way the reduction of the model bias due to the implementation of the multiphase chemistry. For instance, on an annual mean, the initial HCOOH column bias of the model standard simulation ($-1.97 \pm 1.64 \times 10^{15}$ molecules cm^{-2}) is reduced to -0.88 ± 1.62 and $0.99 \pm 2.16 \times 10^{15}$ molecules cm^{-2} with the EMAC_(dih) and EMAC_(diol) simulations, respectively (Fig. 1). On a seasonal basis, such a significant bias reduction can be observed throughout the year (ED Figs 3,4), though it appears that the model implementing the multiphase chemistry tends to overpredict the HCOOH columns in September-November in the tropics, especially over the Amazon Basin that is simulated too dry and too hot during the dry season (and hence with too much temperature-dependent emissions of hydrocarbon precursors). On this topic, we also refer to the paragraph in the manuscript that is dedicated to the remaining model-to-satellite discrepancies (see here below). We also added statistics to the comparison between the modelled and ground-based FTIR HCOOH columns (ED Fig.

1), which also indicate a significant reduction of the column bias at most FTIR stations. Finally, similar probability histograms as for the HCOOH comparisons have been made to better characterize the HCHO columns comparison between the model simulations and the OMI satellite measurements (ED Fig. 5). On an annual mean, the implementation of the multiphase chemistry does not significantly affect the model-to-OMI comparison, though locally, over specific months, larger changes can be observed (see ED Fig. 8).

“Although the multiphase mechanism fills the gap between model and measurements globally, the $EMAC_{(dioh)}$ and $EMAC_{(diol)}$ simulations over- and under-predict the HCOOH columns over tropical and boreal forests, respectively. We primarily ascribe these remaining discrepancies to inaccuracies in the predicted HCHO distributions as compared to OMI/Aura measurements (ED Fig. 5). Regional under-(over-)estimation of modelled HCHO indeed translates through the multiphase conversion to under-(over-)prediction of HCOOH (ED Fig. 6). For instance, underestimated biomass burning emissions of VOCs lead to under-predicted abundance of HCHO, and hence of HCOOH, such as during the 2010 Russian wildfires (ED Fig. 6a-d). Conversely, the too high model temperatures over Amazonia during the dry season induce an excess in isoprene emissions, which results in too high HCHO and HCOOH levels (ED Fig. 6i-l). More realistic VOC emissions and HCHO modelling will eventually lead to further improvements in predicted HCOOH. On the other hand, the model over-prediction over the tropical forests for HCOOH might also be reduced if the recently discovered reaction with stabilized Criegee intermediates were considered (Vereecken et al., 2017; Carvan et al., 2020). With this additional HCOOH sink, $EMAC_{(diol)}$ would represent the most realistic prediction for this organic acid. Implementation of the α -hydroperoxycarbonyls photolysis (Liu et al., 2018; Müller et al., 2019) and photo-oxidation of aromatics (Wang et al., 2020) might further improve the representation of HCOOH locally.”

I am also curious, why they have chosen to focus exclusively on remote sensing column data. There are many recent European and NASA aircraft experiments that include HCOOH, HCHO, CO, etc that may provide additional constraints on the hypothesis being proposed.

The use of remote sensing column data is motivated by the fact that they provide consistent and extensive global measurements, which allows us to grasp the global situation in a straightforward way. In particular, the IASI and OMI satellite measurements offer - from a single instrument - a daily (quasi-)global coverage of the HCOOH and HCHO distributions, respectively. The ground-based FTIR observations provide a (quasi-) continuous sampling of the atmosphere in the vicinity of the station, with a higher accuracy than the satellite instruments.

Conversely, aircraft observations are relatively sparse in space and time, and hence are representative for a specific region, at a specific time, only. Moreover, despite the measurements of many VOCs, HCOOH concentration is not always measured. For example, to the best of our knowledge, only a few aircraft campaigns took place during the 2010-2012 time period, namely CalNex, DC3 and Discover-

AQ, which mainly focused on continental US only, with sometimes a limited availability in HCOOH measurements.

The authors make a comment about inferring isoprene emissions from HCHO columns using an inverse model. They may well be right with a Bayesian approach. If their hypothesis is correct the additional HCOOH source would be larger during the wet season when there is extensive cloud cover and HCHO cannot be observed using UV wavelengths. I suggest more statistical evidence is produced on a seasonal timescale is produced before they can make a statement about casting doubt on these studies. Even with a 30% reduction in model columns (as they state) due to the additional HCOOH source the model still has a large positive bias compared to OMI (Figure 6 in extended data).

During revision of the manuscript, we noticed a model bug which for the former $EMAC_{(diol)}$ simulation led to HCHO phase transfer being simulated still with the effective Henry's law coefficient. Together with the explicit hydration/dehydration of HCHO, it resulted in an effective HCHO solubility being 1000 times higher than the correct value. Nevertheless, the revised model still predicts a substantial enhancement of HCOOH levels. Following the new model simulations, the impact of the mechanism on predicted HCHO is much reduced, but still significant (by as much as 15-20%) over specific regions and for specific months (see ED Fig. 8). Considering the limited impact on HCHO globally, we no longer suggest that the multiphase chemistry improves the comparison with the OMI data nor that the top-down isoprene emissions can be largely biased. In the manuscript, we only mention that, locally, the top-down emissions estimates can eventually benefit from the implementation of the multiphase chemistry in the model:

“With $EMAC_{(diol)}$, we predict over tropical source regions as much as 15-20% decrease of HCHO columns compared to $EMAC_{(base)}$ during specific months (ED Fig. 8). Note that it does not affect the overall agreement between the model and OMI data (ED Fig. 5). However, we anticipate the estimates of regional isoprene and other hydrocarbons emissions based on HCHO source inversions will be improved once the multiphase conversion of formaldehyde is accounted for.”

Similarly, the authors argue that the additional HCOOH sources helps to improve the agreement with MOPITT and in situ CO measurements. I am unconvinced with the analysis as currently reported. In Figure 7 in the extended data, the dramatic changes in colour are somewhat misleading. A change in yellow to red could simply be a change of 10%. A clear statistical report would go a long way to supporting their case. The comparison with in situ data is even less convincing. Without accompanying statistics I cannot comment further. However, based on the plots provided I suspect reporting meaningful changes in ground-based CO is probably a bit of a stretch.

For the reasons exposed above, we have performed new model simulations in which the impact on CO is much reduced. Therefore, we have removed the comparisons with CO from MOPITT and in situ measurements. We now only mention the reduction in global tropospheric CO yield from methane chemistry (see ED Fig. 9), which brings the global mean CO yield closer to the isotope estimates:

“The $EMAC_{(diol)}$ reduced formaldehyde concentrations also result in lower modelled CO yield from methane oxidation, notably over remote areas where methane oxidation represents the main source of atmospheric CO (Fisher et al., 2015) (ED Fig. 9). On a global scale, the average tropospheric CO yield from methane oxidation changes from 0.91 ($EMAC_{(base)}$) to 0.88-0.90 ($EMAC_{(diol)}$ and $EMAC_{(diox)}$, respectively), in agreement with isotope-enabled inversion estimates (Bergamaschi et al., 2000).”

Referee #2 (Remarks to the Author):

This is a relatively straightforward paper about the atmospheric chemistry of formic acid. Several recent studies have highlighted that formic acid is more abundant in the atmosphere than currently known sources can account for. The paper uses some previously published kinetic data to argue that methanediol, formed from formaldehyde hydrolysis in cloud droplets, can partition to the gas phase, where it is oxidized by OH to form formic acid. The mechanism is included in a global model and the resulting formic acid columns are compared with those from the IASI satellite instrument and from ground-based FTIR column measurements. Including the mechanism gives a better description of the observations, particularly in regions with strong isoprene emissions, and also lowers the atmospheric burdens of carbon monoxide and formaldehyde. The Authors argue that this brings the model results for these species also in better agreement with the CO and formaldehyde observations. Finally, the Authors discuss other atmospheric implications of these findings, including the acidity of cloud and rain water, and the atmospheric burdens of sulfur dioxide and hydrogen peroxide.

I find the paper interesting and important, but have to admit that I am not completely overwhelmed by the evidence. Figure 1 shows that the new mechanism can account for higher formic acid over regions with high biogenic emissions, but the IASI data also show high columns at northern mid latitudes and over the boreal forests in North America and Russia. There is virtually no discussion of this in the paper.

Thank you for considering our research work and for your comprehensive review. Your constructive comments really helped to improve the manuscript and to make the conclusions more robust. As mentioned previously, we have indeed performed chamber experiments whose results, corroborated by theoretical calculations, provide evidence for the new mechanism of HCOOH formation that we propose. Such evidence also strengthens the conclusions of the study.

We acknowledge that there is still an under-prediction of HCOOH in the boreal regions (and an overprediction over Amazonia), as highlighted by both IASI and ground-based FTIR data. We now attribute most of these remaining biases to model deficiencies in representing tropospheric HCHO levels, specifically in these regions (see ED Fig. 6). These deficiencies include, for instance, an underestimation of the biomass burning emissions, which leads to an important under-prediction of HCHO and eventually of HCOOH (as can be observed with the 2010 Russian fires depicted in ED Fig. 6). Conversely, over Amazonia, the model is too dry and overestimates temperatures in the dry season, which leads to too strong isoprene emissions and results in too high HCHO levels (and to a HCOOH overprediction). Of course, we cannot rule out the existence of other missing sources and/or sinks of HCOOH that are not yet implemented in the model. It is now discussed in the manuscript as follows:

“Although the multiphase mechanism fills the gap between model and measurements globally, the $EMAC_{(diox)}$ and $EMAC_{(diol)}$ simulations over- and under-predict the HCOOH columns over tropical and boreal forests, respectively. We primarily ascribe these remaining discrepancies to inaccuracies in the predicted HCHO distributions as compared to OMI/Aura measurements (ED Fig. 5). Regional under-(over-)estimation of modelled HCHO indeed translates through the multiphase conversion to under-(over-)prediction of HCOOH (ED Fig. 6). For instance, underestimated biomass burning emissions of VOCs lead to under-predicted abundance of HCHO, and hence of HCOOH, such as during the 2010 Russian wildfires (ED Fig. 6a-d). Conversely, the too high model temperatures over Amazonia during the dry season induce an excess in isoprene emissions, which results in too high HCHO and HCOOH levels (ED Fig. 6i-l). More realistic VOC emissions and HCHO modelling will eventually lead to further improvements in predicted HCOOH. On the other hand, the model over-prediction over the tropical forests for HCOOH might also be reduced if the recently discovered reaction with stabilized Criegee intermediates were considered (Vereecken et al., 2017; Carvan et al., 2020). With this additional HCOOH sink, $EMAC_{(diol)}$ would represent the most realistic prediction for this organic acid. Implementation of the α -hydroperoxycarbonyls photolysis (Liu et al., 2018; Müller et al., 2019) and photo-oxidation of aromatics (Wang et al., 2020) might further improve the representation of HCOOH locally.”

In addition, given the laboratory facilities available to the Authors, I had expected a new experimental confirmation of the proposed mechanism, or an atmospheric measurement of methanediol. While the Authors discuss the lack of methanediol measurements in the SI, there was no effort to characterize how existing instruments respond to this compound.

Thank you for encouraging us to provide experimental evidence of the multiphase chemical mechanism; it really helps to strengthen the manuscript. We have now been able to perform simulation experiments at the SAPHIR atmospheric chamber (Forschungszentrum Jülich) and to acquire experimental evidence of methanediol formation consistent with our proposed mechanism. The main pieces of information obtained from these experiments are:

- efficient outgassing of methanediol from formaldehyde dissolved in liquid droplets
- a constraint on the reactivity of methanediol in the gas phase
- a unity-yield production of formic acid from methanediol oxidation by OH in the gas phase

The results of the experiment are summarized in the main revised manuscript (Fig. 3) and detailed in the Supplementary Information (SI Sect.1 and 4).

Having said that, the proposed mechanism does provide a potential and new explanation for a long-standing problem in atmospheric chemistry, and the publication of this paper would spur a lot of new activity to evaluate these findings.

We very much appreciate this positive judgement of our work.

Detailed comments:

The manuscript does not devote much attention to the role of nitrogen oxides. One issue with formic acid has been that several of the known formation mechanisms are most efficient in low NO_x conditions, whereas efficient formic acid formation has also been observed at high NO_x. I encourage the Authors to consider this aspect of their work in more detail.

The multiphase formation mechanism we present here is not directly dependent on NO_x. It is well known that HCHO-yield from VOC oxidation is higher under high-NO_x conditions. Combined with elevated levels of OH in polluted regions, both precursor formation and transformation to formic acid could be expected to show the largest efficiency, at least on a time scale of a day. Nevertheless, the model predicts no clear dependence on NO_x on a monthly mean basis. This can be seen in Fig. R2 (here below) where the yield of formic acid from the scavenged formaldehyde is plotted against NO_x for the boreal summer. We now mention in the main manuscript that the production of formic acid is mostly independent of the NO_x concentration:

“The additional HCOOH production, essentially independent of NO_x, allows the model predictions to reach the measured HCOOH levels derived from IASI and to reduce significantly the mean model-to-satellite biases from $-1.97(\pm 1.64) \times 10^{15}$ (EMAC_(base)) to $-0.88(\pm 1.62) \times 10^{15}$ (EMAC_(dloh)) and $0.99(\pm 2.16) \times 10^{15}$ (EMAC_(diol)) molecules cm⁻² (Fig. 1).”

Figure R1. Scatter plot of the predicted yield of formic acid ($P(\text{HCOOH})_{\text{diol}}$) from the scavenged formaldehyde ($L(\text{HCHO})_{\text{scav}}$) as a function of NO_x mixing ratio for the $\text{EMAC}_{(\text{diol})}$ simulation. Model daily mean output is shown for the boreal summer 2010 (June - August) filtered for $P(\text{HCOOH})_{\text{diol}} > 1e^{-15}$ mol/mol/s, $L(\text{HCHO})_{\text{scav}} > 10^{-15}$ mol/mol/s, and NO_x from 10 to 10^4 pmol/mol.

In several instances, I believe that the paper could do more to explore alternative explanations of its findings. For example, biomass burning provides the largest direct emission source of formic acid and these emissions are difficult to describe accurately in models. There is barely any mention of this source and its uncertain emissions in the paper. Could the high formic acid columns observed by IASI over the boreal forests in the summer be due to wildfires?

Indeed, the uncertainties of biomass burning emission modelling are large. The emission factors also vary due to the transition from the flaming to the smoldering stage (e.g., Zheng et al., 2018). We have already conducted a sensitivity study by changing the emission factors for HCOOH by Akagi et al. (2011) to the older by Andreae and Merlet (2001), which are a factor of 6 higher for extratropical forests (0.54 and 2.9 g/kg, respectively). Nevertheless, the changes on the HCOOH budget were almost negligible, especially compared to the new multiphase source. Direct HCOOH global production from biomass burning in EMAC amounts to 2.5 Tg/yr only (Table 1 in the manuscript). Despite the uncertainties on fire emissions, it is unlikely that direct HCOOH emissions from biomass burning is a significant contributor. Paulot et al. (2011), Stavrakou et al. (2012) (applying inverse modelling), and Millet et al. (2015) all concluded that the large missing source of HCOOH is more consistent with secondary formation rather than missing primary emissions. Furthermore, Chaliyakunnel et al. (2016) provided evidence of a quick, massive secondary production of HCOOH in biomass burning plumes, and ruled out the direct emissions of HCOOH from fires as being the main missing source: “If a 10-fold bias were to extend to fires in other regions, biomass burning could produce 14 Tg/a of HCOOH in the

tropics or 16 Tg/a worldwide. However, even such an increase would only represent 15–20% of the total required HCOOH source, implying the existence of other larger missing sources.”

We rather believe that biomass burning may play a role because of the photochemical production from the many VOC precursors that are emitted by fires. Franco et al. (2020) have shown that the HCOOH enhancements captured by IASI in summer over, e.g., boreal forests, are due to both biogenic and biomass burning emissions. This enhanced photochemical production due to fires also involves the multiphase pathway we are presenting here. Indeed, due to the important amount of HCHO that originates (directly and indirectly) from fires, additional HCOOH is also produced indirectly from biomass burning via the outgassing of methanediol from HCHO-enriched droplets and its subsequent OH-oxidation. In ED Fig. 6, we show that locally large model under-(over-)prediction of HCHO as to OMI leads to large under-(over-)prediction of HCOOH as to IASI. The first example illustrates the situation of the 2010 Russian wildfires (ED Fig. 6a-d), and shows that EMAC largely underpredicts the HCHO levels and hence the HCOOH abundance as well. Therefore, we are convinced that deficiencies and inaccuracies in the biomass burning emissions used by the model result in a mismatch between the modelled and observed patterns of HCHO and HCOOH, and that it explains part of the remaining discrepancies between EMAC and IASI, especially over the boreal regions.

It is now discussed in the manuscript, along with other potential missing sources of HCOOH that might further improve the model simulations. Please see our answer to your first comment.

Likewise, the paper is quick to conclude that lower formation sources of CO and formaldehyde in the model are more consistent with the measurements, but there are many other reasons that could account for these differences, including uncertain emissions of CO and of the biogenic precursors of CO and formaldehyde, oxidant concentrations, simplified chemical mechanisms in global models, etc.

During revision of the manuscript, we noticed a model bug which for the former EMAC_(diol) simulation led to HCHO phase transfer being simulated still with the effective Henry's law coefficient. Together with the explicit hydration/dehydration of HCHO, it resulted in an effective HCHO solubility being 1000 times higher than the correct value. Nevertheless, the revised model still predicts a substantial enhancement of HCOOH levels. Following the new model simulations, the impact of the mechanism on predicted HCHO is much reduced, but still significant (by as much as 15-20%) over specific regions and for specific months (see ED Fig. 8). Considering the limited impact on HCHO globally, we no longer suggest that the multiphase chemistry improves the comparison with the OMI data nor that the top-down isoprene emissions can be largely biased. In the manuscript, we only mention that, locally, the top-down emissions estimates can eventually benefit from the implementation of the multiphase chemistry in the model. For the same reason, the impact on CO is much reduced. Therefore, we have removed the comparisons with CO from MOPITT and in situ measurements. We now only mention

the reduction in global tropospheric CO yield from methane chemistry (see ED Fig. 9), which brings the global mean CO yield closer to the isotope estimates. This part of the manuscript now reads:

“With $EMAC_{(diol)}$, we predict over tropical source regions as much as 15-20% decrease of HCHO columns compared to $EMAC_{(base)}$ during specific months (ED Fig. 8). Note that it does not affect the overall agreement between the model and OMI data (ED Fig. 5). However, we anticipate the estimates of regional isoprene and other hydrocarbons emissions based on HCHO source inversions will be improved once the multiphase conversion of formaldehyde is accounted for. The $EMAC_{(diol)}$ reduced formaldehyde concentrations also result in lower modelled CO yield from methane oxidation, notably over remote areas where methane oxidation represents the main source of atmospheric CO (Fisher et al., 2015) (ED Fig. 9). On a global scale, the average tropospheric CO yield from methane oxidation changes from 0.91 ($EMAC_{(base)}$) to 0.88-0.90 ($EMAC_{(diol)}$ and $EMAC_{(dih)}$, respectively), in agreement with isotope-enabled inversion estimates (Bergamaschi et al., 2000).”

As to the CO and HCHO differences, it is worth mentioning that our model includes over 40 primarily emitted VOC species which are degraded in a gas-phase chemical mechanism nearing the 2000 reactions-mark. The levels and dynamics of oxidant concentrations predicted with our model have been presented in Lelieveld et al. (2016). We are not aware of any other global model having this level of details. Thus, we currently think that the bias in modelled emissions is likely the main reason for the CO and HCHO differences. For instance, ED Fig. 6 presents a few examples of significant mismatch between the modelled and OMI HCHO columns over a few regions, for specific months. From these examples, it is clear that deficiencies in the modelled emissions (e.g., biomass burning emissions, such as shown with the 2010 Russian wildfires; ED Fig. 6a-d) lead to large under- or overprediction of the HCHO levels (and hence of the levels of other species).

It is interesting that the mechanism can also account for acetic acid formation from acetaldehyde. In this case, however, acetaldehyde measurements have been higher than global models can account for, and this issue would presumably only get worse if the acetaldehyde to acetic acid pathway were included in a global model. Some discussion of this would be good to add.

We now provide both experimental evidence and theoretical calculations of the multiphase production of formic acid. However, we still lack similar data for acetic acid as well as a realistic source of acetaldehyde in our global model EMAC. Therefore, we decided not to implement the relevant processes in the new set of simulations we use for the revised manuscript. We are currently investigating the sources of both acetaldehyde and acetic acid in the framework of a separate study.

Figure 1: Are the formic acid columns shown in panels A-C actually comparing the same thing? As discussed in the SI, the satellite instrument is more sensitive to the mid troposphere than to the

surface. A brief discussion in the main body of the paper of the averaging kernel of the instrument, and to what extent this has, or has not, been taken into account in the comparison with the model output would help to better understand and appreciate this graph.

As mentioned in the introduction to our replies, we now exploit a new satellite product for HCOOH, obtained with a neural network-based retrieval approach. Specifically, the retrieval method consists in quantifying, for each IASI observation, the amplitude of the HCOOH signal, and converting it directly into HCOOH total column using an artificial feedforward neural network (NN) (Franco et al., 2018). This NN is fed by other IASI observed parameters to account explicitly for the state of the surface and atmosphere, and assumes a vertical profile of HCOOH to convert the spectral signal into total column. The NN does not rely directly on a priori information (conversely to optimal estimation-like retrieval methods), hence no averaging kernels are produced. Indeed, though the IASI sensitivity to HCOOH is maximal in the mid-troposphere and decreases outside that range (Pommier et al., 2016), the NN is informed of the state of the atmosphere (via the temperature and humidity profiles from IASI) and therefore can deduce how much each atmospheric layer contributes to the signal as observed by IASI. The NN has been trained to account for this non-homogenous vertical signal, and to produce a HCOOH total column that can be compared directly with other total column data e.g., from global models. The disadvantage of this approach is that a fixed vertical profile shape has to be assumed. Fortunately, the associated uncertainty that is introduced is modest, as explained below.

Regarding the retrieval uncertainties, the estimated uncertainty on each input variable is propagated through the NN to yield an uncertainty on each single-pixel retrieved column (Franco et al., 2018). For a typical non-background HCOOH abundance ($0.3\text{--}2.0 \times 10^{15}$ molecules cm^{-2}), the relative uncertainty on an individual retrieved column ranges from 10 to 50%, with the highest uncertainties found for the low columns. This uncertainty increases for lower background columns. However, these uncertainties mostly cancel out by averaging multiple IASI measurements on the EMAC spatial grid. For instance, around 17 satellite measurements are averaged per day on one single $\sim 1.8^\circ \times 1.8^\circ$ model grid box at the Equator, which corresponds to $>18,000$ measurements over 2010–2012. This number increases with the latitude and the higher spatial sampling of IASI due to its polar orbit. The main source of uncertainty is actually the shape of the HCOOH vertical profile that is assumed in the retrieval. Franco et al. (2018) evaluated this uncertainty to be of the order of 20% by comparing two estimates of the true HCOOH columns obtained in turn by two NNs assuming a completely different shape of the profile. A comparison with independent HCOOH columns from ground-based FTIR measurements at various latitudes and environments did not reveal any large systematic biases of the IASI data (Franco et al., 2020). We estimate remaining biases to be an order of magnitude lower than the initial model underprediction ($\text{EMAC}_{(\text{base})}$) of the HCOOH columns and therefore sufficiently accurate to demonstrate the large improvements brought with the new mechanism. In the Methods, we provide a summary of the retrieval and of the main information on the new IASI data.

Please note that the HCOOH total columns derived from the ground-based FTIR observations are retrieved using an optimal estimation-based method. Therefore, in that case averaging kernels are produced and applied subsequently to smooth the model profiles for the evaluation of the model simulations (SI Sect. 6). The random and systematic uncertainties affecting the retrieved total columns are in the range of 11–13 and 15–18%, respectively. As with the model-to-satellite comparison, these

uncertainties are much smaller than the initial model underprediction of the FTIR HCOOH columns (ED Fig. 1), and hence do not affect any of the conclusions of the study.

Referee #3 (Remarks to the Author):

Review of “Ubiquitous atmospheric production of organic acids mediated by warm clouds” by Franco, et al. for publication in Nature:

This is a highly intriguing manuscript suggesting that the hydration of formaldehyde within cloud-water to form methylene glycol (HOCH₂OH, methanediol) and subsequent evaporation and gas-phase oxidation thereof by OH radical forming formic acid to be an important, or rather, the dominant source (more than 4 times greater than all other known sources combined) of formic acid to the atmosphere. The authors suggest this chemistry changes our understanding of the pH in clouds and rain water, particularly in the tropical regions, through significant increases in acidity due to higher levels of formic acid produced by this mechanism (conversion of formaldehyde to formic acid). The authors also suggest similar chemistry involving larger aldehydes may be important for organic aerosol formation and growth.

Understanding the levels of organic acids, particularly the abundant formic and acetic acids in the atmosphere has been a long-standing problem in the field of atmospheric chemistry. Here the authors of the present work present a hypothesis which can more than explain the observed abundance of formic acid in the atmosphere. While the hypothesis presented here is indeed interesting, logical and plausible, and worthy of continued work, and even publishable as a hypothesis-type paper within a more specialized journal, it is not clear that this manuscript, as it currently stands, meets the criteria for publication in a high profile type of journal such as Nature, due primarily to the large uncertainties inherent in the methods used to validate the hypothesis – uncertainties, which in my opinion, preclude quantitative assessment of the importance of the suggested chemistry for the atmosphere. These uncertainties are discussed further below.

Thank you very much for considering our research work and for your constructive review. Your detailed comments helped greatly in improving the manuscript and encouraged us to perform additional works in order to strengthen our main conclusions.

As explained in the introduction to our answers, we do provide now the experimental evidence on the methanediol outgassing from HCHO-enriched droplets and on its subsequent production of formic acid upon oxidation by OH. Moreover, it is supported by theoretical calculations. This experimental and theoretical evidence is, on its own, an original finding and, we believe, a substantial addition that reinforces the study. It is now presented in the new version of the paper and fully detailed in Supplementary Information Sections 1 and 2. We have performed new model simulations that implement the rate constants derived from the experiment.

Henry's Law Constant constant of HOCH₂OH: The authors use a value of $1.0 \times 10^4 \text{ M atm}^{-1}$ which is obtained from the computer software HENRYWIN provided by the US EPA using a bond or group specific structure additivity relationship (SAR) to calculate the water solubility of specific molecules. While this software produces solubility results in reasonable agreement with experimental results for some species, it also produces quite unreliable results for others. Therefore, one should assign a high uncertainty toward results from this SAR, when no experimental values exist (i.e., it is not inconceivable that the Henry's Law constant for this species could be of order 50 times greater than the number used within these simulations). For instance, the bond method from HENRYWIN calculates ethylene glycol solubility to be about 80 times smaller than recommended by the JPL Kinetics Panel [Burkholder, et al. 2015]. A similar error for methylene glycol solubility would result in the same increase in the ratio of methylene glycol in the condensed to that in the gas phase, and presumably have an important impact on the amount of formic acid produced from the subsequent gas-phase HOCH₂OH oxidation. The authors clearly need to discuss the uncertainty in this quantity and the sensitivity of their results toward this.

We agree about the uncertainty of the Henry's law constant for methanediol. Therefore, we have conducted an additional simulation (called $\text{EMAC}_{(\text{dih})}$ in the manuscript) with a value 100 times greater (i.e., 10^6 M atm^{-1}). The results are now presented in the main manuscript and, along with the results from the $\text{EMAC}_{(\text{diol})}$ simulation assuming a low solubility, provide a range of additional HCOOH that is produced through the multiphase pathway. Despite the much higher solubility of methanediol, $\text{EMAC}_{(\text{dih})}$ still predicts a substantial amount of HCOOH ($\sim 45 \text{ Tg yr}^{-1}$) that is bigger than all the previously-known sources combined (see Table 1 in the manuscript and ED Table 1), which has never been reported before in the literature. In the different model comparisons to the IASI and ground-based FTIR columns, it is clear that $\text{EMAC}_{(\text{dih})}$ already significantly reduces the gap between model and observations. Therefore, we can fairly consider that $\text{EMAC}_{(\text{dih})}$ predictions represent the lower range of additional HCOOH that can be produced via the multiphase pathway, whereas predictions from $\text{EMAC}_{(\text{diol})}$ can be considered as the upper range.

Depositional loss of HOCH₂OH: The authors do not discuss the importance of dry (and wet) deposition for methylene glycol. Instead only the 'rapid gas-phase oxidation [by OH]' is discussed. Though, the assumed OH rate constant, $1.3 \times 10^{-11} \text{ cm}^3 \text{ molec}^{-1} \text{ s}^{-1}$, yields an average OH lifetime of methylene glycol of ~ 21 hours for $[\text{OH}] = 1 \times 10^6 \text{ molec cm}^{-3}$. This is comparable to dry deposition timescale within the terrestrial boundary layer. Inclusion of this process may be important.

We have indeed included all the standard loss processes for methanediol. Cloud scavenging, wet deposition, dry deposition over oceans and continents are represented. We apologize for not having made this clear before. We now describe some aspects of dry deposition modelling in Section 3.a.iii of the SI. The lifetime of methanediol against dry deposition over vegetation in the $\text{EMAC}_{(\text{dih})}$

simulation ($H(\text{HOCH}_2\text{OH}) = 1\text{e}^6 \text{ M/atm}$) is predicted to be often in the range 1-5 days on an annual mean basis (see Fig. R2 here below). This is comparable to the sink by reaction with OH.

Figure R2. Comparison between the sink by dry deposition and the one by reaction with OH for methanediol (HOCH_2OH) with the assumption of $H=10^6 \text{ M/atm}$. The lower panels show the lifetimes against these sinks on an annual mean basis.

Gas phase dehydration of $\text{HOCH}_2\text{OH}(\text{g})$: While it is well-established that the barrier to un-catalyzed dehydration of methylene glycol in the gas phase is large ($\sim 43 \text{ kcal/mol}$), precluding this reaction from being important under atmospherically relevant temperatures, calculations have indicated that this process can be catalyzed by a number of other species [e.g., Kumar and Francisco, 2015, and references therein]. Given the scarcity of atmospheric work on this species it is surprising the authors did not include a discussion this paper and references therein in this manuscript. Investigation of the potential importance of catalyzed dehydration of methylene glycol is necessary to determine the robustness of the author's assumption that $\text{OH} + \text{methylene glycol}(\text{g})$ will be the dominant fate of this species within the atmosphere.

The work of Kumar and Francisco (2015) does shed doubts whether reaction with OH is the only gas-phase sink for HOCH_2OH . The authors computed only the energy barriers with and without methanediol being complexed. Fortunately, Kumar et al. (2017) have recently reported theoretical

calculations for the effective rate constants of HOCH₂OH-complex decomposition. The highest computed rate constant is for the complex with HCOOH being about 10^{-10} s^{-1} which is much lower than the estimated OH sink ($\sim 10^{-5} \text{ s}^{-1}$). This is fully consistent with the experimental and theoretical evidence we have obtained as can be seen in SI Fig.1 (reported here below). In the upper right panel, the HOCH₂OH signal (attributed to the difference between the HCHO measurements with the Hantzsch and with DOAS instrument) is well reproduced by the box model after the CO injection (pink shaded area) that suppressed the oxidation by OH. The only remaining sinks, which are accounted for in the box model, are heterogeneous loss to the walls ($8.5 \times 10^{-5} \text{ s}^{-1}$) and dilution by synthetic air ($\sim 3\%/hour$) to compensate for the leakages from the teflon walls of the SAPHIR chamber.

Supplementary Information Figure 1. Time series of measured HCHO, HOCH₂OH, HCOOH and OH (blue dots) together with results from chemical box modelling (red lines) during an experiment in the simulation outdoor chamber SAPHIR. Grey shaded areas indicate conditions without sunlight and pink areas indicate times when excess CO was present as OH scavenger.

We now explicitly refer to the experimental and theoretical results in the manuscript:

“Here we provide evidence that methanediol reaction with OH in the gas phase quantitatively yields HCOOH under atmospheric conditions (Fig. 2). By conducting experiments with the atmospheric simulation chamber SAPHIR (SI Sect. 1), we indeed show that formaldehyde in aqueous solution is efficiently converted to methanediol immediately after injection and quantitatively yields HCOOH upon photo-oxidation (Fig. 3). This is supported by theoretical calculations (SI Sect. 2).”

OH + HOCH₂OH rate constant uncertainty: Needed to address both of the previous comments is the potential uncertainty in the assumed OH + methylene glycol rate constant. This rate coefficient has not been measured and is simply calculated here using an SAR which has no validation for gem-diol type functional groups. This is likely uncertain by a factor of 2-3.

We agree. As shown in the SI Fig. 1 (reported above) and SI Sect. 1.b,c, the experimental data on the photo-oxidation of methanediol is consistent with a rate constant $k = 7.5 \times 10^{-12} \text{ cm}^3 \text{ s}^{-1}$, which is only ~25% lower than the one estimated with the SAR in the first version of the manuscript. We now use this value in the box model as well as in the global model.

An experimental study on the gas phase OH oxidation chemistry (kinetics and products) of methylene glycol would go a long way toward reducing the uncertainties discussed above, and in my opinion, is a needed prerequisite to publish this work in a journal such as Nature.

We would like to thank the reviewer again for encouraging us to pursue an experimental study. Therefore, as explained above, we put much effort into designing and conducting experiments with the SAPHIR chamber. In addition, the findings are supported by theoretical calculations that have been performed in parallel. We are convinced that it removes most of the uncertainties on the atmospheric fate of methanediol and we hope it addresses the main concerns on this point.

The authors use comparisons of formic acid column abundance from chemical transport model simulations and remote sensing observations (ground and satellite spectroscopic measurements) to help support their hypothesis. As stated above, it is clear that simulations using traditional chemistry substantially under-predicts formic acid in the atmosphere. Simulations using the proposed mechanism, however, substantially over-predict formic acid in many regions (particularly in the tropics) relative to the observations. Perhaps uncertainties in assumptions may play a role here? In addition, the spatial correspondence between the model and satellite column measurements is not particularly convincing, especially in the summertime over 40N-80N region (figure 1). What is the reason behind this? It may be helpful to perform a 'tagged' formic acid analysis where formic acid from various sources is tracked separately, enabling the parsing of discrepancies as a function of source.

As for the modelling uncertainties: As reported above, we have obtained experimental and theoretical evidence that helps into reducing part of the uncertainties related to the modelling of methanediol fate. Moreover, we have addressed and ameliorated the Amazon dry bias of the EMAC model in the dry season which led to strong isoprene emissions (driven by very high temperatures) and shut down of dry deposition (due to too low soil moisture). For this, we updated the soil moisture stress factor for stomatal conductance and added a more realistic formulation of the non-stomatal dry deposition mainly affecting soluble species (see SI Sect. 3.a.iii). A complete solution of the Amazon dry bias in the

model is beyond the scope of this manuscript. Nevertheless, the remaining overestimation of the total HCOOH column over the Amazon might be further reduced when considering the recently discovered gas-phase sink of HCOOH by reaction with stabilized Criegee Intermediates (Vereecken et al., 2017; Caravan et al., 2020).

As for the spatial correspondence: We fully agree that the spatial correspondence between the model and satellite data is currently not optimal. We primarily attribute it to the deficiencies of the model in reproducing the HCHO global distributions as derived from the OMI satellite measurements. In ED Fig. 6, we show that a local model under- (over-)prediction of HCHO as to the OMI data, leads the model to under-(over-)predict the HCOOH columns compared to the IASI measurements. Therefore, we believe that the HCHO under-prediction in the summertime over the 40-80° N region is the main reason behind the too low HCOOH columns that are modelled in this region. Nevertheless, we cannot rule out that other sources of modelling uncertainties are also at play (e.g., the HCOOH reaction with SCIs that is mentioned above). We are now convinced that further improvement of the HCOOH modelling can be achieved by addressing these issues, in particular the modelled HCHO misrepresentation in some environments.

These uncertainties are now discussed in the manuscript, along with other potential missing sources of HCOOH that might further improve the model simulations, as follows:

“Although the multiphase mechanism fills the gap between model and measurements globally, the $EMAC_{(dioh)}$ and $EMAC_{(diol)}$ simulations over- and under-predict the HCOOH columns over tropical and boreal forests, respectively. We primarily ascribe these remaining discrepancies to inaccuracies in the predicted HCHO distributions as compared to OMI/Aura measurements (ED Fig. 5). Regional under-(over-)estimation of modelled HCHO indeed translates through the multiphase conversion to under-(over-)prediction of HCOOH (ED Fig. 6). For instance, underestimated biomass burning emissions of VOCs lead to under-predicted abundance of HCHO, and hence of HCOOH, such as during the 2010 Russian wildfires (ED Fig. 6a-d). Conversely, the too high model temperatures over Amazonia during the dry season induce an excess in isoprene emissions, which results in too high HCHO and HCOOH levels (ED Fig. 6i-l). More realistic VOC emissions and HCHO modelling will eventually lead to further improvements in predicted HCOOH. On the other hand, the model over-prediction over the tropical forests for HCOOH might also be reduced if the recently discovered reaction with stabilized Criegee intermediates were considered (Vereecken et al., 2017; Carvan et al., 2020). With this additional HCOOH sink, $EMAC_{(diol)}$ would represent the most realistic prediction for this organic acid. Implementation of the α -hydroperoxycarbonyls photolysis (Liu et al., 2018; Müller et al., 2019) and photo-oxidation of aromatics (Wang et al., 2020) might further improve the representation of HCOOH locally.”

Therefore, we believe that performing a HCOOH-tagged simulation would be too demanding in the framework of the present study, and would not provide significant additional insights, especially compared to the benefits of further improvements of the HCHO modelling.

With regard to comparisons of model simulations using the proposed mechanism of formic acid with NDACC station observations: In the remote regions the model often predicts substantially higher formic acid than is observed by the ground-based FTIR measurements. A robust understanding of these differences in the remote areas may be fruitful, as the chemistry should be much simpler. Also, the seasonal shape is often somewhat different between model and observations. Are these things understood? Why is the model not averaged over the same time period as the observations? This can and should be done. Why is Wollongong averaging kernel treated differently?

Regarding the model overprediction as to the remote NDACC stations: In this revised version of the study, we exploit a new set of model simulations that provide a range of predicted HCOOH columns. This entire range is lower than the HCOOH columns that were produced in the initial version of the paper. Therefore, we do not systematically overpredict the observed HCOOH abundance at remote NDACC stations, as can be seen in Extended Data Fig. 1.

As to the new comparison with the FTIR data, we agree that the model tends to overpredict the low HCOOH columns and to under-predict the high columns, and hence to exhibit a more “flattened” seasonal cycle. This is a typical pattern that is observed when comparing global models to FTIR data. It is mainly due to the artificial dilution of the model information over relatively large areas (~200 x 200 km model grid boxes), whereas the FTIR observations are representative for a much smaller environment. A good example is the Jungfraujoch station, located on the Northern crest of the Swiss Alps (at ~3.5 km a.s.l), whereas the model pixel including that station also encompasses the industrialized and populated southwestern part of Germany. For the same reason, local enhancements due for instance to biomass burning events (e.g., Wollongong) or city emissions (e.g., Toronto) can hardly be captured by the relatively coarse resolution of the model predictions. Moreover, it is likely that the underlying emissions implemented in the model are not peaked enough to accurately represent local emissions enhancements. The motivation behind the model-to-FTIR comparison was to assess in one grasp the ability of the multiphase production of HCOOH at reducing the gap between model and FTIR measurements recorded at various latitudes, in various environments.

The reason why we considered initially a longer time period for the FTIR measurements was to compensate for the sparsity of the FTIR measurements due, e.g., to bad weather conditions or instrumental failures. In this revised manuscript, the FTIR observations are now averaged over the same time period as the model data (i.e. 2010-2012).

The retrieval method applied to the Wollongong FTIR spectra simply does not produce averaging kernels.

In addition, the authors use comparisons of model simulations of HCHO and CO against satellite observations to support the hypothesis of the paper. However, these are fairly weak constraints, on the proposed mechanism, as modeled HCHO abundance is highly sensitive to VOC emissions, particular oxidation mechanisms, and chemical environment in which the oxidation occurs (i.e., NO levels), all of which are uncertain in the model realm. The changes in CO between simulations is small, and not well-distinguishing as compared with the data.

During revision of the manuscript, we noticed a model bug which for the former EMAC_(diol) simulation led to HCHO phase transfer being simulated still with the effective Henry's law coefficient. Together with the explicit hydration/dehydration of HCHO, it resulted in an effective HCHO solubility being 1000 times higher than the correct value. Nevertheless, the revised model still predicts a substantial enhancement of HCOOH levels. Following the new model simulations, the impact of the mechanism on predicted HCHO is much reduced, but still significant (by as much as 15-20%) over specific regions and for specific months (see ED Fig. 8). Considering the limited impact on HCHO globally, we no longer suggest that the multiphase chemistry improves the comparison with the OMI data nor that the top-down isoprene emissions can be largely biased. In the manuscript, we only mention that, locally, the top-down emissions estimates can eventually benefit from the implementation of the multiphase chemistry in the model. For the same reason, the impact on CO is much reduced. Therefore, we have removed the comparisons with CO from MOPITT and in situ measurements. We now only mention the reduction in global tropospheric CO yield from methane chemistry (see ED Fig. 9), which brings the global mean CO yield closer to the isotope estimates. This part of the manuscript now reads:

“With EMAC_(diol), we predict over tropical source regions as much as 15-20% decrease of HCHO columns compared to EMAC_(base) during specific months (ED Fig. 8). Note that it does not affect the overall agreement between the model and OMI data (ED Fig. 5). However, we anticipate the estimates of regional isoprene and other hydrocarbons emissions based on HCHO source inversions will be improved once the multiphase conversion of formaldehyde is accounted for. The EMAC_(diol) reduced formaldehyde concentrations also result in lower modelled CO yield from methane oxidation, notably over remote areas where methane oxidation represents the main source of atmospheric CO (Fisher et al., 2015) (ED Fig. 9). On a global scale, the average tropospheric CO yield from methane oxidation changes from 0.91 (EMAC_(base)) to 0.88-0.90 (EMAC_(diol) and EMAC_(diox), respectively), in agreement with isotope-enabled inversion estimates (Bergamaschi et al., 2000).”

The multiphase formation mechanism we present here is not directly dependent on NO_x. It is well known that HCHO-yield from VOC oxidation is highest under high-NO_x conditions. Combined with elevated levels of OH in polluted regions, both the precursor formation and transformation to formic acid could be expected to show the largest efficiency, at least on a time scale of a day. Nevertheless, the model predicts no clear dependence on NO_x on a monthly mean basis. This can be seen in Fig. R1 (here below) where the yield of formic acid from the scavenged formaldehyde is plotted against NO_x

for the boreal summer. We now mention in the main manuscript that the formic acid production is mostly independent of the NO_x concentration:

“The additional HCOOH production, essentially independent of NO_x, allows the model predictions to reach the measured HCOOH levels derived from IASI and to reduce significantly the mean model-to-satellite biases from $-1.97(\pm 1.64)\times 10^{15}$ (EMAC_(base)) to $-0.88(\pm 1.62)\times 10^{15}$ (EMAC_(dih)) and $0.99(\pm 2.16)\times 10^{15}$ (EMAC_(diol)) molecules cm⁻² (Fig. 1).”

Figure R1. Scatter plot of the predicted yield of formic acid (P(HCOOH)_{diol}) from the scavenged formaldehyde (L(HCHO)_{scav}) as a function of NO_x mixing ratio for the EMAC_(diol) simulation. Model daily mean output is shown for the boreal summer 2010 (June - August) filtered for P(HCOOH)_{diol} > 1e⁻¹⁵ mol/mol/s, L(HCHO)_{scav} > 10⁻¹⁵ mol/mol/s, and NO_x from 10 to 10⁴ pmol/mol.

Finally, it has been observed that model/observation discrepancies between formic and acetic acids are often both similar in magnitude and correlated [Paulot, et al, 2009]. Is this coincidence? The hypothesis put forward here only helps to resolve the formic acid discrepancy. How does one make sense of this?

Recently, Franco et al. (2020) have reported the first global distributions of acetic acid obtained from spaceborne measurements. They have shown that formic and acetic acid enhancements are highly correlated over source regions, and that those are mainly driven similarly by biogenic and/or biomass burning emissions, depending on the regions. However, over the same source regions, they highlighted some discrepancies between the two organic acids suggesting that formation pathways

specific to each species can also be at play. In the new set of simulations we have decided not to consider the analogous multiphase reactions for acetaldehyde leading to acetic acid anymore, since we lack the experimental evidence we obtained for HCOOH. However, from the literature values of the hydration/dehydration kinetics for acetaldehyde, we think that this pathway is unlikely to produce acetic acid in a magnitude similar to the one of HCOOH. Given that the atmospheric sources of acetaldehyde are strongly underestimated, we still think that this multiphase pathway could be a significant contributor to atmospheric acetic acid. The sources of acetaldehyde and acetic acid will be the subject of a separate study in the future.

References:

- Akagi, S. K. *et al.* Emission factors for open and domestic biomass burning for use in atmospheric models. *Atmos. Chem. Phys.*, 11(9), 4039–4072, 2011. <https://doi.org/10.5194/acp-11-4039-2011>
- Andreae, M. O. and Merlet, P. Emission of trace gases and aerosols from biomass burning. *Global Biogeochem. Cy.*, 15(4), 955–966, 2001. <https://doi.org/10.1029/2000GB001382>
- Bergamaschi, P., Hein, R., Brenninkmeijer, C. A. M. & Crutzen, P. J. Inverse modeling of the global CO cycle: 2. Inversion of $^{13}\text{C}/^{12}\text{C}$ and $^{18}\text{O}/^{16}\text{O}$ isotope ratios. *J. Geophys. Res. - Atmos.*, 105, 1929–1945, 2000. <https://doi.org/10.1029/1999JD900819>
- Boyce, S. D. and Hoffmann, M. R. Kinetics and mechanism of the formation of hydroxymethanesulfonic acid at low pH. *J. Phys. Chem. A*, 88, 4740–4746, 1984. <https://doi.org/10.1021/j150664a059>
- Burkholder, J. B. *et al.* Chemical Kinetics and Photochemical Data for Use in Atmospheric Studies, Evaluation No. 18, JPL Publication 15-10, Jet Propulsion Laboratory, Pasadena, 2015 <http://jpldataeval.jpl.nasa.gov>.
- Caravan, R. L. *et al.* Direct kinetic measurements and theoretical predictions of an isoprene-derived Criegee intermediate. *Proc. Natl. Acad. Sci. U.S.A.*, 117, 9733–9740, 2020. <https://doi.org/10.1073/pnas.1916711117>
- Chaliyakunnel, S. *et al.* A Large Underestimate of Formic Acid from Tropical Fires: Constraints from Space-Borne Measurements. *Environ. Sci. Technol.*, 50, 11, 5631–5640, 2016. <https://doi.org/10.1021/acs.est.5b06385>
- Fisher, J. A. *et al.* Seasonal changes in the tropospheric carbon monoxide profile over the remote Southern Hemisphere evaluated using multi-model simulations and aircraft observations. *Atmos. Chem. Phys.*, 15, 3217–3239, 2015. <https://doi.org/10.5194/acp-15-3217-2015>
- Franco, B. *et al.* A general framework for global retrievals of trace gases from IASI: Application to methanol, formic acid, and PAN. *J. Geophys. Res. - Atmos.*, 123, 13,963–13,984, 2018. <https://doi.org/10.1029/2018JD029633>
- Franco, B. *et al.* Spaceborne measurements of formic and acetic acids: A global view of the regional sources. *Geophys. Res. Lett.*, 47, e2019GL086239, 2020. <https://doi.org/10.1029/2019GL086239>
- Funderburk, L. H. *et al.* Mechanisms of general acid and base catalysis of the reactions of water and alcohols with formaldehyde. *J. Am. Chem. Soc.*, 100, 5444–5459, 1978. <https://doi.org/10.1021/ja00485a032>
- Jöckel, P. *et al.* Development cycle 2 of the Modular Earth Submodel System (MESSy2). *Geosci. Model Dev.*, 3, 717–752, 2010. <https://doi.org/10.5194/gmd-3-717-2010>
- Jöckel, P. *et al.* Earth System Chemistry integrated Modelling (ESCiMo) with the Modular Earth Submodel System (MESSy) version 2.51. *Geosci. Model Dev.*, 9, 1153–1200, 2016. <https://doi.org/10.5194/gmd-9-1153-2016>

- Kumar, M. and Francisco, J. S. The role of catalysis in alkanediol decomposition: Implications for general detection of alkanediols and their formation in the atmosphere. *J. Phys. Chem. A*, 119, 9821–9833, 2015. <https://doi.org/10.1021/acs.jpca.5b07642>
- Kumar, M., *et al.* Role of Proton Tunneling and Metal-Free Organocatalysis in the Decomposition of Methanediol: A Theoretical Study. *J. Phys. Chem. A*, 121, 22, 4318–4325, 2017. <https://doi.org/10.1021/acs.jpca.7b01864>
- Lelieveld, J. *et al.* Global tropospheric hydroxyl distribution, budget and reactivity. *Atmos. Chem. Phys.*, 16, 12477–12493, 2016. <https://doi.org/10.5194/acp-16-12477-2016>
- Liu, Z. *et al.* The photolysis of α -hydroperoxycarbonyls. *Phys. Chem. Chem. Phys.*, 20, 6970–6979, 2018. <https://doi.org/10.1039/C7CP08421H>
- Millet, D. B. *et al.* A large and ubiquitous source of atmospheric formic acid. *Atmos. Chem. Phys.*, 15 (11), 6283–6304, 2015. <https://doi.org/10.5194/acp-15-6283-2015>
- Müller, J.-F. *et al.* Chemistry and deposition in the Model of Atmospheric composition at Global and Regional scales using Inversion Techniques for Trace gas Emissions (MAGRITTE v1.1) Part 1: Chemical mechanism. *Geosci. Model Dev.*, 12, 2307–2356, 2019. <https://doi.org/10.5194/gmd-12-2307-2019>
- Paulot, F. *et al.* Importance of secondary sources in the atmospheric budgets of formic and acetic acids. *Atmos. Chem. Phys.*, 11, 1989–2013, 2011. <https://doi.org/10.5194/acp-11-1989-2011>
- Pommier, M. *et al.* HCOOH distributions from IASI for 2008–2014: comparison with ground-based FTIR measurements and a global chemistry-transport model. *Atmos. Chem. Phys.*, 16, 8963–8981, 2016. <https://doi.org/10.5194/acp-16-8963-2016>
- Stavrakou, T. *et al.* Satellite evidence for a large source of formic acid from boreal and tropical forests. *Nat. Geosci.*, 5 (1), 26–30, 2012. <https://doi.org/10.1038/ngeo1354>
- Vereecken, L. *et al.* Unimolecular decay strongly limits the atmospheric impact of Criegee intermediates. *Phys. Chem. Chem. Phys.*, 19, 31599–31612, 2017. <https://doi.org/10.1039/C7CP05541B>
- Wang, S. *et al.* Aromatic Photo-oxidation, A New Source of Atmospheric Acidity. *Environ. Sci. Technol.*, 54, 7798–7806, 2020. <https://doi.org/10.1021/acs.est.0c00526>
- Winkelman, J. G. M. *et al.* Kinetics and chemical equilibrium of the hydration of formaldehyde. *Chem. Eng. Sci.*, 57, 4067–4076, 2002. [https://doi.org/10.1016/S0009-2509\(02\)00358-5](https://doi.org/10.1016/S0009-2509(02)00358-5)
- Zheng, B. *et al.* On the role of the flaming to smoldering transition in the seasonal cycle of African fire emissions. *Geophys. Res. Lett.*, 45, 11,998–12,007, 2018. <https://doi.org/10.1029/2018gl079092>

Reviewer Reports on the First Revision:**Ref #1**

First, my apologies for the delay in the second review. I am not going to repeat the summary of key results etc. I have now had a chance to re-read the paper, the reviews and the author responses and the revised supplementary information.

The paper has been substantially revised. It's the same broom but with a new head and a new handle. Certainly, my comments have been addressed by the authors and I believe the manuscript has been revised to incorporate the new information provided. The statistical analysis is much better and supports the original claims. I am appreciative of the new laboratory data collected at SAPHIR, which makes the paper much stronger in my opinion. The authors have also taken advantage of the long delay in their revision to use an improved dataset.

I do believe it is worth publishing in Nature. It addresses a long standing puzzle in atmospheric chemistry that has been partly addressed by confronting models with satellite data.

Ref #2

N/A

Ref #3

The authors have done quite a nice job addressing the reviewer comments from the first round or review. I think this brings the manuscript much closer to being publishable in Nature. The chamber experiments do substantially strengthen the paper. However, there are still a few remaining issues which should be addressed before I can recommend publication. I do encourage the authors to consider these seriously, as shown through their simulations tests, even with reduced magnitudes, this process still remains of dominant importance in the production of formic acid in the atmosphere. Highlighting the most realistic understanding within the manuscript will certainly boost the long-term impact of this work.

Major comments:

1) Upon further consideration, the Henry's law value for methylene diol (from HenryWin prediction software) appears to be anomalously low. While the authors have conducted simulations using, in my opinion, a more realistic value (factor of 100 greater, $1 \times 10^6 \text{ M atm}^{-1}$) the revised manuscript remains largely skewed towards presentation of the results using the lower value for Henry's Law constant. I suggest revising this presentation to reflect a more reasonable KH value. Using measured values of KH for similar species one may arrive at such an estimate. For example, by comparing pairs of alcohols and hydroperoxides with the same carbon number (methanol/methyl hydrogen peroxide, ethanol/ethyl hydrogen peroxide, acetic acid/peroxy acetic acid) it can be inferred that the hydroperoxide and alcohol groups contribute similarly to solubility, despite the difference of an additional O atom in the peroxide species. Given this, and the fact that the HOCH₂OOH species has measured KH(298K) of $1.7 \times 10^6 \text{ M atm}^{-1}$, one may infer the $1 \times 10^6 \text{ M atm}^{-1}$ value (K_{dioh}) to be much closer to the truth than the result produced by the HenryWin software. The statistical comparisons from Figure 1 appear to support this conjecture. I will also note, the model may be missing other HC(O)OH sources, so underprediction of HC(O)OH by the model may not be unphysical in the same sense as overprediction (e.g., something is missing in the modeled boreal regions).

2) Another issue apparently missed during the first review and revision, is the fact that solubility is

a very strong function of temperature. No discussion of the T-dependence for the solubility of methylene diol (or the other species) is given in the paper. For instance, using a temperature coefficient of 8500 K^{-1} (average of observed values for H_2O_2 and HOCH_2OOH) with $K(\text{H}) = A \cdot \exp(8500/\text{K}) \text{ M atm}^{-1}$, one can estimate that KH will change by a factor of ~ 80 over the 260-300K range, with higher values at lower temperatures. Suggest T-dependent KH simulation is needed, if this has not been done. This may improve the correspondence of model/satellite $\text{HC}(\text{O})\text{OH}$ maps, something which as-is, is not completely convincing.

3) What about the same process occurring from aerosol liquid water? This could be a much more distributed/ubiquitous source of $\text{HC}(\text{O})\text{OH}$ (e.g., the diffuse background)?

Specific comments:

LN107: I'm not sure stating a range in solubility is the best here without further explanation. At minimum, the specific value used in the simulations should be noted here, in the main paper.

LN117-119: Agree the formation of $\text{HC}(\text{O})\text{OH}$ via diol formation/gas-phase oxidation should be NO_x independent, however the underlying HCHO formation IS likely NO_x dependent, and the extent to which the model does not get this correct (it is known models tend not to reproduce observed NO_x well) may bias the inferred $\text{HC}(\text{O})\text{OH}$ formation. This was the point of the question in the first review. Is it possible to constrain or nudge the model HCHO using observed HCHO, and bypass such uncertainties?

Ref #4

The paper is a revision of a much earlier submission from 2017, which I also reviewed. In comparison with that earlier submission, the Authors have performed new chamber experiments that demonstrate formic acid formation from methanediol released from an aerosolized formaldehyde solution. This addition addresses one of the main limitations of the earlier submission. The Authors uncovered an error in their earlier model calculation: the solubility of formaldehyde was 1000 times higher than the correct value. After correcting for this error, the global formic acid production from the proposed mechanism is much reduced, but still large enough that it does not change the main message of the paper. The effect of the new multiphase chemistry mechanism on the formaldehyde and CO budgets is now also much reduced and no longer deemed to be large enough to be supported by measurements. Finally, the Authors have updated the formic acid columns from IASI using a new, neural-network based algorithm, which has been shown elsewhere to result in improved formic acid columns. Overall, I believe that most of my original comments were addressed in a satisfactory manner and that the manuscript has been significantly improved. Some new comments are below.

A. Efficient formic acid production has also been observed in clear air (Yuan et al., 2015), in which case the multiphase mechanism described here cannot be the explanation. I believe this should be acknowledged and discussed. Could this chemistry also happen in aerosol water?

B. In the last paragraph, the Authors raise the possibility that the multiphase mechanism described here may also play a role in the formation of other organic acids. One has to wonder if the Authors can strengthen the paper by doing more with this hypothesis. There is a lot of literature on the formation of acids and aldehydes (work of Kimitaka Kawamura, Patrick Veres, Eric Apel, Rainer Volkamer), and a synthesis of observations could be insightful. Do observations show corresponding organic acids for each of the major aldehydes? This paper focuses on formic acid from formaldehyde, but what do we know about acetaldehyde-acetic acid, propanal-propionic acid, acrolein-acrylic acid, and methacrolein-methacrylic acid observations? The solubility and rate coefficients are probably different enough for each aldehyde-acid pair that one would not expect a quantitative correspondence, but just the composition could be very interesting to look into.

Minor comments:

Line 35: Suggest “nucleation” instead of “activation”.

Lines 69-80: This discussion is very vague in terms of the specific mechanisms that produce formic acid studied in the literature, and could be improved.

Figure 3 caption: It would be helpful if the caption, or the description of the figure in the main text, would have some detail about the measurement techniques that are used to generate these results. Methanediol measurements are not standard observations and the reader should not be required to read the SI to find out how these results were obtained.

Lines 136-138: Please specify what “recently discovered reaction with stabilized Criegee intermediates” you refer to. This is a very vague statement.

Section 4 of the SI: The question is raised how the PTR-TOF-MS responds to methanediol but it seems as if the Authors themselves should have the answers. According to Section 1 of the SI, you used a PTR-TOF-MS for the experiments shown in Figure 3 of the main text. Was there a detectable signal at the parent ion mass of methanediol? I agree that methanediol may be detected at the same product ion as formaldehyde, but since the time series of methanediol and formaldehyde are so different in these experiments, it seems that the experiments should still hold information about the detection of methanediol by PTR-TOF-MS.

References

Yuan, B., Veres, P. R., Warneke, C., Roberts, J. M., Gilman, J. B., Koss, A. R., et al. (2015). Investigation of secondary formation of formic acid: urban environment vs. oil and gas producing region. *Atmospheric Chemistry and Physics*, 15, 1975–1993.

Author Rebuttals to First Revision:

We thank all the referees for their positive evaluation of our revised version of the study. Please find here below the point-by-point reply (in blue) to the comments.

Sincerely,

Bruno Franco and Domenico Taraborrelli

On behalf of all co-authors.

Referee #1 (Remarks to the Author):

First, my apologies for the delay in the second review. I am not going to repeat the summary of key results etc. I have now had a chance to re-read the paper, the reviews and the author responses and the revised supplementary information.

The paper has been substantially revised. It's the same broom but with a new head and a new handle. Certainly, my comments have been addressed by the authors and I believe the manuscript has been revised to incorporate the new information provided. The statistical analysis is much better and supports the original claims. I am appreciative of the new laboratory data collected at SAPHIR, which makes the paper much stronger in my opinion. The authors have also taken advantage of the long delay in their revision to use an improved dataset.

I do believe it is worth publishing in Nature. It addresses a long standing puzzle in atmospheric chemistry that has been partly addressed by confronting models with satellite data.

Thank you very much for reviewing our revised study and for the positive evaluation.

Referee #2 (Remarks to the Author):

The authors have done quite a nice job addressing the reviewer comments from the first round or review. I think this brings the manuscript much closer to being publishable in Nature. The chamber experiments do substantially strengthen the paper. However, there are still a few remaining issues which should be addressed before I can recommend publication. I do encourage the authors to consider these seriously, as shown through their simulations tests, even with reduced magnitudes, this process still remains of dominant importance in the production of formic acid in the atmosphere. Highlighting the most realistic understanding within the manuscript will certainly boost the long-term impact of this work.

Thank you very much for this review. To address the first comment on solubility, we have revised the text and figures to present now both simulations (with upper and lower estimate of the solubility) on complete equal footing. As we explain below, these provide respectively, reasonable lower and upper estimate of the extra formic acid produced via the multiphase pathway. As for the comment on temperature dependence: accurate experimental determination of this dependence and subsequent implementation in the model simulations represent an enormous body of work, and we feel would be much more deserving of dedicated follow-up studies. We agree that a discussion was missing; this has now been remedied. We have addressed the other comments by adapting the text where needed, as detailed below.

Major comments:

1) Upon further consideration, the Henry's law value for methylene diol (from HenryWin prediction software) appears to be anomalously low. While the authors have conducted simulations using, in my opinion, a more realistic value (factor of 100 greater, $1 \times 10^6 \text{ M atm}^{-1}$) the revised manuscript remains largely skewed towards presentation of the results using the lower value for Henry's Law constant. I suggest revising this presentation to reflect a more reasonable KH value. Using measured values of KH for similar species one may arrive at such an estimate. For example, by comparing pairs of alcohols and hydroperoxides with the same carbon number (methanol/methyl hydrogen peroxide, ethanol/ethyl hydrogen peroxide, acetic acid/peroxy acetic acid) it can be inferred that the hydroperoxide and alcohol groups contribute similarly to solubility, despite the difference of an additional O atom in the peroxide species. Given this, and the fact that the HOCH₂OOH species has measured KH(298K) of $1.7 \times 10^6 \text{ M atm}^{-1}$, one may infer the $1 \times 10^6 \text{ M atm}^{-1}$ value (Kdih) to be much closer to the truth than the result produced by the HenryWin software. The statistical comparisons from Figure 1 appear to support this conjecture. I will also note, the model may be missing other HC(O)OH sources, so underprediction of HC(O)OH by the model may not be unphysical in the same sense as overprediction (e.g., something is missing in the modeled boreal regions).

Due to the lack of experimental data to constrain methanediol solubility at any temperature, we used the HENRYWIN v3.10 software. It provides, using three methods, estimates of the Henry's law coefficient for methanediol ($\text{H}_{\text{HOCH}_2\text{OH}}$) spanning almost two orders of magnitude ($0.25\text{--}9.3 \times 10^4 \text{ M atm}^{-1}$). The bond method in particular provides reliable estimates of solubility for alcohols and other compounds similar to methanediol within a factor of 3 (the experimental data are from Sander et al., 2011):

	Experimental data	HENRYWIN v3.10
CH ₃ OH	203	234
CH ₃ OOH	294	147
HOCH ₂ OH	/	~1 x 10 ⁴
HOCH ₂ OOH	1.7 x 10 ⁶	~4 x 10 ⁶
CH ₃ CH ₂ OH	193	176
CH ₃ CH ₂ OOH	334	110
CH ₃ C(O)OH	4.8 x 10 ³	1.8 x 10 ³
CH ₃ C(O)OOH	8.3 x 10 ²	7.2 x 10 ²

Hence, we can fairly assume that it provides a good estimate of $H_{\text{HOCH}_2\text{OH}}$ as well. Therefore, we used the value obtained via the bond method ($1.015 \times 10^4 \text{ M atm}^{-1}$), which is the intermediate HENRYWIN result, to run the $\text{EMAC}_{(\text{diol})}$ simulation. To account for the uncertainties on $H_{\text{HOCH}_2\text{OH}}$, we performed another simulation ($\text{EMAC}_{(\text{dih})}$), in which $H_{\text{HOCH}_2\text{OH}}$ is set two orders of magnitude higher (10^6 M atm^{-1}). This cumulates both the uncertainty around the intermediate result of HENRYWIN (one order of magnitude) and the possible temperature dependence of $H_{\text{HOCH}_2\text{OH}}$. From similar compounds, we estimated that the latter can enhance the solubility by roughly one order of magnitude at the typical warm cloud temperatures (please see our reply to the next comment). For these reasons, we considered $H_{\text{HOCH}_2\text{OH}}=10^6 \text{ M atm}^{-1}$ ($\text{EMAC}_{(\text{dih})}$) and $H_{\text{HOCH}_2\text{OH}}=10^4 \text{ M atm}^{-1}$ ($\text{EMAC}_{(\text{diol})}$) as, respectively, the upper and lower estimate of methanediol solubility. In absence of experimental constraints, we believe this is a reasonable approach.

We acknowledge that our approach was not clearly described in the documents. Therefore, we now introduce more faithfully these two simulations in the manuscript (see the text here below) and the Methods. We also explain the rationale behind our approach in the SI Sect. 3.b.iii.

“We implemented in EMAC the explicit kinetic model for the aqueous-phase transformations and the bi-directional phase transfer of methanediol. The solubility of methanediol is not known at any temperature and its estimates span two orders of magnitude (SI Sect. 3.b.iii). We gauge the impact of this uncertainty on the results by performing the simulations $\text{EMAC}_{(\text{diol})}$ and $\text{EMAC}_{(\text{dih})}$ with a lower ($\sim 10^4 \text{ M atm}^{-1}$) and an upper estimate ($\sim 10^6 \text{ M atm}^{-1}$) of Henry’s law constant (solubility), respectively (Methods).”

In addition, we have added the next statement to the manuscript (in the paragraph dedicated to the HCOOH budget) on how the $\text{EMAC}_{(\text{dih})}$ and $\text{EMAC}_{(\text{diol})}$ results must be interpreted:

“ $\text{EMAC}_{(\text{dih})}$ and $\text{EMAC}_{(\text{diol})}$ provide, respectively, a lower and an upper estimate of the extra HCOOH produced via the multiphase processing of formaldehyde.”

Previously, more attention was given to $\text{EMAC}_{(\text{diol})}$, but we agree that this was not the best approach. Therefore, in the manuscript we now discuss both $\text{EMAC}_{(\text{dih})}$ and $\text{EMAC}_{(\text{diol})}$ results in a balanced way. Please note that the main results were already displayed for both simulations (Fig. 1 and Table 1). Several figures in Extended Data have also been updated to include $\text{EMAC}_{(\text{dih})}$. When it was not possible due to a lack of space, $\text{EMAC}_{(\text{diol})}$ has been displayed (because of the larger magnitude of its results which are easier to interpret) and in the figure caption we refer to the corresponding $\text{EMAC}_{(\text{dih})}$ figure in SI Sect. 8.

2) Another issue apparently missed during the first review and revision, is the fact that solubility is a very strong function of temperature. No discussion of the T-dependence for the solubility of methylene diol (or the other species) is given in the paper. For instance, using a temperature

coefficient of 8500 K^{-1} (average of observed values for H_2O_2 and HOCH_2OOH) with $K(\text{H}) = A \cdot \exp(8500/\text{K}) \text{ M atm}^{-1}$, one can estimate that $K(\text{H})$ will change by a factor of ~ 80 over the 260-300K range, with higher values at lower temperatures. Suggest T-dependent $K(\text{H})$ simulation is needed, if this has not been done. This may improve the correspondence of model/satellite $\text{HC}(\text{O})\text{OH}$ maps, something which as-is, is not completely convincing.

We agree that the temperature dependence of $H_{\text{HOCH}_2\text{OH}}$ needs to be discussed, which was not done before. However, no experimental data exists to constrain it and the effect of temperature on the solubility of VOCs is complex. Indeed, while the Henry's law constants of many VOCs are usually measured at temperatures not lower than 273K (Table 5-4 of NASA-JPL Evaluation 19-5, <https://jpldataeval.jpl.nasa.gov/pdf/NASA-JPL%20Evaluation%2019-5.pdf>), measurements at temperatures down to $\sim 248\text{K}$ indicate that in supercooled water (such as in the clouds) the solubility of some VOCs decreases with the temperature as much as it increases between 273K and 298K due to ice-like clusters expelling the VOCs to the gas phase (Sieg et al., 2009). Implementing explicitly such a complex dependence for more accurate modelling of methanediol, and evaluating its effects on the multiphase HCOOH production in dedicated model simulations, would first require specific experiments to derive a robust temperature dependence also valid for temperatures below 273K. These experiments are challenging to set up considering the difficulty to measure methanediol in the gas and aqueous phases, as we have experienced with the SAPHIR chamber. Hence, we consider this to be a research topic that should be carefully investigated in the coming years, but that falls outside the scope of the current study. In this regard, as explained in our previous answer, we evaluated that the $H_{\text{HOCH}_2\text{OH}}$ increase due to lower cloud temperatures is roughly of one order of magnitude. This is encompassed in the two orders of magnitude of difference between the $H_{\text{HOCH}_2\text{OH}}$ values we used for $\text{EMAC}_{(\text{dioh})}$ and $\text{EMAC}_{(\text{diol})}$. Therefore, we can reasonably consider that the difference of HCOOH production between the two simulations provides a conservative estimate of the impact of such temperature dependence.

In the manuscript, we now attract the reader's attention on the uncertainties related to the temperature dependence (please, see the modified text sentences in our previous answer), as well as on the need to account for it as follows:

"Implementation of the α -hydroperoxycarbonyls photolysis (Liu et al., 2018; Müller et al., 2019) and photo-oxidation of aromatics (Wang et al., 2020), and of a temperature-dependent solubility for methanediol, would further improve the HCOOH representation."

Beyond the temperature dependence of $H_{\text{HOCH}_2\text{OH}}$, we already present in this study other sources of uncertainties that can explain the remaining biases between the model predictions and the satellite observations. In particular, we discuss the too low VOC emissions (inducing too low formaldehyde and HCOOH levels) over the boreal regions, and a high temperature bias in the model over Amazonia which induces an excess in hydrocarbon emissions and too high VOC concentrations.

3) What about the same process occurring from aerosol liquid water? This could be a much more distributed/ubiquitous source of $\text{HC}(\text{O})\text{OH}$ (e.g., the diffuse background)?

Indeed, it will be interesting to evaluate the impact of this multiphase mechanism in aerosol liquid water. We already plan to extend our multiphase mechanism to deliquescent aerosols. The model framework is relatively similar to the one implemented for cloud droplets, but it is currently under testing and non-linear effects, like the one of ionic strength on kinetics, must still be represented. We expect that our proposed mechanism will result in a non-negligible HCOOH production in polluted regions (i.e., with an elevated aerosol content), close to the VOC sources (i.e., with high formaldehyde

levels), and that it will help to improve the model predictions in cloud-free conditions. In this regard, we have added the following sentence to the last paragraph of the manuscript:

“We showed that this multiphase pathway involving aldehyde hydrates is decisive in predicting organic acid formation and atmospheric acidity. It could also be significant in the presence of deliquescent aerosols and would explain the elevated HCOOH levels in cloud-free conditions (Yuan et al., 2015).”

Specific comments:

LN107: I'm not sure stating a range in solubility is the best here without further explanation. At minimum, the specific value used in the simulations should be noted here, in the main paper.

We agree. As explained above, we now discuss in detail in the Supplementary Information Sect. 3.b.iii. the current uncertainties on the Henry's law coefficient of methanediol and we justify the two values that were selected to run the EMAC_(dih) and EMAC_(diol) simulations. We have therefore changed these lines in the manuscript, as indicated above, to mention these uncertainties and to provide the values that were used. In addition, we refer to the SI Sect. 3.b.iii. for the detailed discussion.

LN117-119: Agree the formation of HC(O)OH via diol formation/gas-phase oxidation should be NO_x independent, however the underlying HCHO formation IS likely NO_x dependent, and the extent to which the model does not get this correct (it is known models tend not to reproduce observed NO_x well) may bias the inferred HC(O)OH formation. This was the point of the question in the first review. Is it possible to constrain or nudge the model HCHO using observed HCHO, and bypass such uncertainties?

Indeed, the HCHO formation is NO_x-dependent, and representing such a dependency well in global models is challenging. We have already explained in the manuscript (lines 126-136) that inaccuracies in the predicted HCHO (compared to satellite measurements) is likely a major factor that impedes the model to better match the satellite HCOOH distributions. Hence, the misrepresentation of the NO_x contributes to biases in the modelled HCHO, which in turn results in biases in the subsequent HCOOH formation. We now attract the attention of the reader to that point in lines 135-136 as follows:

“More realistic VOC emissions, and enhanced modelling of formaldehyde and its NO_x-dependence, will eventually lead to further improvements in predicted HCOOH.”

Using satellite HCHO to nudge the model simulations of HCOOH is an interesting idea. In chemistry global modelling, this is routinely performed for long-lived species (typically methane or carbon dioxide), but we are not aware of similar nudging involving satellite observations of reactive species such as HCHO. It is difficult to assess what it implies. In principle, it can be done with the EMAC model (it is done for long-lived tracers). However, the satellite retrievals of HCHO are not vertically resolved and only produce an integrated column. This implies that, somehow, to nudge a model, one would need to assume a vertical distribution of HCHO, which is extremely variable in space and time. That would propagate to the model predictions the assumptions related to the vertical distribution and diurnal cycle of HCHO, as well as the uncertainties associated with the satellite measurements.

References:

- Liu, Z., Nguyen, V. S., Harvey, J., Müller, J.-F. & Peeters, J. The photolysis of α -hydroperoxycarbonyls. *Physical Chemistry Chemical Physics* **20**, 6970–6979 (2018).
- Müller, J.-F., Stavrou, T. & Peeters, J. Chemistry and deposition in the Model of Atmospheric composition at Global and Regional scales using Inversion Techniques for Trace gas Emissions (MAGRITTE v1.1). Part 1: Chemical mechanism. *Geosci. Model Dev.* **12**, 2307–2356 (2019).
- Sander, S. P., et al.: Chemical Kinetics and Photochemical Data for Use in Atmospheric Studies, Evaluation No. 17, JPL Publication 10-6, Jet Propulsion Laboratory, Pasadena, <http://jpldataeval.jpl.nasa.gov>, 2011.
- Sieg, K., Starokozhev, E., Schmidt, M. U. & Püttmann, W. Inverse temperature dependence of Henry's law coefficients for volatile organic compounds in supercooled water. *Chemosphere* **77**, 8-14 (2009).
- Yuan, B., et al.: Investigation of secondary formation of formic acid: urban environment vs. oil and gas producing region, *Atmos. Chem. Phys.*, **15**, 1975–1993, <https://doi.org/10.5194/acp-15-1975-2015>, 2015.
- Wang, S. *et al.* Aromatic Photo-oxidation, A New Source of Atmospheric Acidity. *Environmental Science & Technology* **54**, 7798–7806 (2020).

Referee #3 (Remarks to the Author):

The paper is a revision of a much earlier submission from 2017, which I also reviewed. In comparison with that earlier submission, the Authors have performed new chamber experiments that demonstrate formic acid formation from methanediol released from an aerosolized formaldehyde solution. This addition addresses one of the main limitations of the earlier submission. The Authors uncovered an error in their earlier model calculation: the solubility of formaldehyde was 1000 times higher than the correct value. After correcting for this error, the global formic acid production from the proposed mechanism is much reduced, but still large enough that it does not change the main message of the paper. The effect of the new multiphase chemistry mechanism on the formaldehyde and CO budgets is now also much reduced and no longer deemed to be large enough to be supported by measurements. Finally, the Authors have updated the formic acid columns from IASI using a new, neural-network based algorithm, which has been shown elsewhere to result in improved formic acid columns. Overall, I believe that most of my original comments were addressed in a satisfactory manner and that the manuscript has been significantly improved. Some new comments are below.

Thank you very much for your comprehensive assessment, positive evaluation and further comments.

A. Efficient formic acid production has also been observed in clear air (Yuan et al., 2015), in which case the multiphase mechanism described here cannot be the explanation. I believe this should be acknowledged and discussed. Could this chemistry also happen in aerosol water?

The possibility that the multiphase production of HCOOH also occurs via aerosol liquid water has also been raised by another referee. Therefore, we repeat here below the same answer to his/her comment.

Indeed, it will be interesting to evaluate the impact of this multiphase mechanism in aerosol liquid water. We already plan to extend our multiphase mechanism to deliquescent aerosols. The framework is relatively similar to the one implemented for cloud droplets, but it is currently under testing and non-linear effects, like the one of ionic strength on kinetics, must still be represented. We expect that our proposed mechanism will result in a non-negligible HCOOH production in polluted regions (i.e., with an elevated aerosol content), close to the VOC sources (i.e., with high formaldehyde levels), and that it will help to improve the model predictions in cloud-free conditions. In this regard, we have added the following sentence to the last paragraph of the manuscript:

“We showed that this multiphase pathway involving aldehyde hydrates is decisive in predicting organic acid formation and atmospheric acidity. It could also be significant in the presence of deliquescent aerosols and would explain the elevated HCOOH levels in cloud-free conditions (Yuan et al., 2015).”

B. In the last paragraph, the Authors raise the possibility that the multiphase mechanism described here may also play a role in the formation of other organic acids. One has to wonder if the Authors can strengthen the paper by doing more with this hypothesis. There is a lot of literature on the formation of acids and aldehydes (work of Kimitaka Kawamura, Patrick Veres, Eric Apel, Rainer Volkamer), and a synthesis of observations could be insightful. Do observations show corresponding organic acids for each of the major aldehydes? This paper focuses on formic acid from formaldehyde, but what do we know about acetaldehyde-acetic acid, propanal-propionic acid, acrolein-acrylic acid, and methacrolein-methacrylic acid observations? The solubility and rate coefficients are probably different enough for each aldehyde-acid pair that one would not expect a quantitative correspondence, but just the composition could be very interesting to look into.

This is a very interesting point. We are currently exploring the formation of other organic acids via the multiphase chemistry of higher aldehydes. Based on the similarities of the hydration/dehydration equilibrium between formaldehyde, glyoxal and methylglyoxal (Doussin and Monod, 2013), we expect a significant formation of glyoxalic acid (and thus oxalic acid) and pyruvic acid. On the other hand, the same equilibrium for acetaldehyde is less favourable, hence we expect a more limited production of acetic acid via this pathway. It is of course a very interesting point to discuss, but currently the efficient formation of higher organic acids remains partly speculative as there are still large uncertainties on the Henry's law coefficients and kinetics of these hydrated aldehydes (diols and polyols). A major difficulty is indeed the lack of experimental data to constrain the Henry's law coefficients for these diols and polyols. Furthermore, measurements of higher aldehydes and acids are sparse and are not always done simultaneously. For these reasons, we limit our discussion in the manuscript by adding the following text on production of organic acids from the most promising glyoxal and methylglyoxal:

“Given the favourable hydration equilibrium constants for major C₂-C₃ carbonyls (Doussin and Monod, 2013), this pathway opens avenues for more realistic representation of other major organic acids, and hence of cloud droplet nucleation and cloud evolution. Especially, we expect the multiphase processing for glyoxal and methylglyoxal to be significant for explaining the observed concentrations of oxalic and pyruvic acids (Yu, 2000).”

Minor comments:

Line 35: Suggest “nucleation” instead of “activation”.

It is indeed more accurate. We have changed “activation” to “nucleation”.

Lines 69-80: This discussion is very vague in terms of the specific mechanisms that produce formic acid studied in the literature, and could be improved.

We agree that this introductory paragraph does not give much information on the specific mechanisms. Unfortunately, the chemical pathways that lead - or that have been proposed to lead - to the formation of atmospheric HCOOH are many, various and not easily explained. In view of the overall length of the document, it is therefore not possible to be more specific on these mechanisms. However, most of them are either described in the Methods section or in the Supplementary Information, or are referred to in the manuscript. For instance, the usually admitted formation pathways that are implemented in EMAC are described in the Methods section and in the SI Sect. 3. In the manuscript (lines 140-142), we also refer to two mechanisms that have been proposed very recently, which might further improve the representation of atmospheric HCOOH: the photolysis of α -hydroperoxycarbonyls (produced by the isoprene oxidation), which would yield $\sim 5 \text{ Tg yr}^{-1}$ of HCOOH (Müller et al., 2019), and the oxidation of intermediates produced by the photo-oxidation of aromatics, which nonetheless has not been tested yet with global modelling (Wang et al., 2020). Regarding other mechanisms that have already been tested in global models (but were not confirmed or were shown not to yield HCOOH), they include, for example, the OH-initiated oxidation of isoprene and of its degradation products hydroxyacetone and glycolaldehyde, the OH-oxidation of monoterpenes, the degradation of isoprene nitrates, the reaction of methyl peroxy radical (CH₃O₂) with OH and the subsequent Criegee intermediate + H₂O reaction, and a diffuse source of HCOOH from the aging of organic aerosols (e.g., Paulot et al., 2011; Millet et al., 2015). In the Methods section, we mention most of them and why they were not implemented in EMAC.

Figure 3 caption: It would be helpful if the caption, or the description of the figure in the main text, would have some detail about the measurement techniques that are used to generate these results. Methanediol measurements are not standard observations and the reader should not be required to read the SI to find out how these results were obtained.

We have added more information on the measurement techniques in the caption of the figure to make it more self-consistent for the reader. The caption now reads:

“Figure 3 | Multiphase production of formic acid in the SAPHIR chamber. Formaldehyde (HCHO) was measured by differential optical absorption spectroscopy (DOAS; in black), while the sum of HCHO and methanediol (HOCH₂OH) was measured with the Hantzsch method. The difference between the Hantzsch and DOAS signals allows visualizing HOCH₂OH (in blue). The formic acid (HCOOH) mixing ratio was monitored by proton transfer reaction – mass spectrometry (PTR-MS) measurements (in red). Upon injection of the formalin (stabilized formaldehyde) solution into the Teflon chamber, HOCH₂OH immediately outgasses from the droplets. The chamber roof is initially closed (grey shaded area; Stage I). Gas-phase HCHO mixing ratio is initially very low but increases to be as abundant as HOCH₂OH just before the start of the photo-oxidation when the roof is opened (yellow shaded area; Stage II). The decay of the HCHO and HOCH₂OH signals is concurrent with an additional production of HCOOH. Finally, addition of carbon monoxide (CO) as an OH-scavenger allowed quantification of the wall effects (Stage III). Detailed experimental results and measurement information are provided in SI Sects 1 and 4.”

Lines 136-138: Please specify what “recently discovered reaction with stabilized Criegee intermediates” you refer to. This is a very vague statement.

Indeed, our statement was not precise enough. We have clarified it as follows:

“Fast reaction of HCOOH with stabilized Criegee intermediates have recently been emphasized (Vereecken et al., 2017; Caravan et al., 2020). The HCOOH over-prediction over the tropical forests might be reduced if this additional sink were considered. ”

Section 4 of the SI: The question is raised how the PTR-TOF-MS responds to methanediol but it seems as if the Authors themselves should have the answers. According to Section 1 of the SI, you used a PTR-TOF-MS for the experiments shown in Figure 3 of the main text. Was there a detectable signal at the parent ion mass of methanediol? I agree that methanediol may be detected at the same product ion as formaldehyde, but since the time series of methanediol and formaldehyde are so different in these experiments, it seems that the experiments should still hold information about the detection of methanediol by PTR-TOF-MS.

We agree with the referee and we acknowledge that, after having done the experiments, we now have this kind of information. We have evidence that methanediol contributes significantly to the PTR-TOF-MS signal detected at 31 m/z as a result of water loss by the parent ion at 49 m/z. Both signals present a time profile that is clearly a combination of formaldehyde and methanediol. With respect to the parent ion mass of methanediol the signal intensity is 2-3 orders of magnitude lower than 31 m/z. Accordingly, we have clarified the text in SI Section 4 by adding:

“Alcohols, when protonated in the PTR-MS instruments, are well known to eliminate a water molecule and methanediol is expected to significantly contribute to m/z 31 in air masses with a cloud processing history. In fact, our chamber experiments yielded a PTR-MS signal at m/z 31 with a time profile that is a combination of the ones for HCHO (DOAS) and HOCH₂OH (Hantzsch - DOAS). This is also the case for the much weaker signal at m/z 49. Our model predictions with EMAC_(dloh) and EMAC_(diol) for the

minimum and maximum interference of methanediol on field measurements of formaldehyde are given as the annual average $\text{HOCH}_2\text{OH}_{(g)}/\text{HCHO}_{(g)}$ ratio (SI Fig. 6)."

References

Yuan, B., Veres, P. R., Warneke, C., Roberts, J. M., Gilman, J. B., Koss, A. R., et al. (2015). Investigation of secondary formation of formic acid: urban environment vs. oil and gas producing region. *Atmospheric Chemistry and Physics*, 15, 1975–1993.

References:

- Caravan, R. L. *et al.* Direct kinetic measurements and theoretical predictions of an isoprene-derived Criegee intermediate. *PNAS* **117**, 9733–9740 (2020).
- Doussin, J.-F. & Monod, A. Structure-activity relationship for the estimation of OH-oxidation rate constants of carbonyl compounds in the aqueous phase. *Atmospheric Chemistry and Physics* **13**, 11625–11641 (2013).
- Millet, D. B. *et al.* A large and ubiquitous source of atmospheric formic acid. *Atmospheric Chemistry and Physics* **15**, 6283–6304 (2015).
- Müller, J.-F., Stavrou, T. & Peeters, J. Chemistry and deposition in the Model of Atmospheric composition at Global and Regional scales using Inversion Techniques for Trace gas Emissions (MAGRITTE v1.1). Part 1: Chemical mechanism. *Geosci. Model Dev.* **12**, 2307–2356 (2019).
- Paulot, F. *et al.* Importance of secondary sources in the atmospheric budgets of formic and acetic acids. *Atmospheric Chemistry and Physics* **11**, 1989–2013 (2011).
- Vereecken, L., Novelli, A. & Taraborrelli, D. Unimolecular decay strongly limits the atmospheric impact of Criegee intermediates. *Physical Chemistry Chemical Physics* **19**, 31599–31612 (2017).
- Wang, S. *et al.* Aromatic Photo-oxidation, A New Source of Atmospheric Acidity. *Environmental Science & Technology* **54**, 7798–7806 (2020).
- Yu, S. Role of organic acids (formic, acetic, pyruvic and oxalic) in the formation of cloud condensation nuclei (CCN): a review. *Atmospheric Research* **53**, 185–217 (2000).

Reviewer Reports on the Second Revision:

Ref #1

N/A

Ref #2

N/A

Ref #3

I maintain that it is implausible that KHdiol is ~400 times lower the measured KH(HMHP). In addition, the authors argument that the HENRYWIN software reproduces observed KH within a factor of three is misleading. As pointed out in the original review for certain glycols it misses the experimental values by more than a factor of 50. Without knowing details of how the HENRYWIN software calculates KH (this is mostly a black box) and upon which species the model was developed/trained, it is impossible to estimate the uncertainty in the calculated KH for an unknown species.

As the experimental values of KHdiol are not available in any published form, I suppose we will need to agree to disagree on the best way to estimate this quantity -- until this quantity has been measured experimentally. However, I think I make a logical argument that KHdiol(298K) of $1e6 \text{ M atm}^{-1}$ is a plausible estimate. In this case, considering reasonable temperature dependencies (even to only 273K) pushes the KHdiol more than an order of magnitude larger than the value termed 'upper limit' in the paper for reduced T. This needs to be acknowledged.

LN 107 add 'at 298K' to end of sentence.

LN 434-436: I do not agree this statement. It can reasonably be estimated that $\text{KH}_{\text{diol}}(298\text{K}) = 1e6 \text{ M/atm}$ (see last review), this along with reasonable temperature coefficient (see last review) yields $\text{KH}_{\text{diol}}(273\text{K}) = 1.4e7 \text{ M/atm}$, and perhaps even higher KH at lower T, though as the authors suggest, things can become complicated as ice begins to form. As this is about 10x greater than upper estimate used, suggest this statement be modified.

SI Sect 1: It is not clear from the language here how the OH + HOCH₂OH reaction coefficient was derived. Early on it says $7.5e-12$ was from fitting MCM simulations to data, while varying the OH + HOCH₂OH rate coefficient in the simulations. Later, experimental rate coefficients of about $2e-11$ are written. Some more information regarding how the two relate and which is the best experimental estimate would be important to add. If the first part is written simply to justify the use of the number in the global simulation, suggest eliminating this and simply state the global simulations were run before the experimental determination, but values are consistent, within uncertainties.

SI P18 last pp: As stated above, the 'factor 3' agreement of HENRYWIN KH with experimentally determined values is misleading. As stated in my first review, HENRYWIN KH for small glycols, arguably molecules much more similar to HOCH₂OH, can differ from experimental values by more than 50x. This statement needs to be corrected.

Ref #4

The authors have addressed my remaining comments in a satisfactory way. I recommend publication of the manuscript.

Author Rebuttals to Second Revision:**Referee #3 (Remarks to the Author):**

Thank you for reviewing our revised study.

I maintain that it is implausible that KHdiol is ~400 times lower the measured KH(HMHP). In addition, the authors argument that the HENRYWIN software reproduces observed KH within a factor of three is misleading. As pointed out in the original review for certain glycols it misses the experimental values

by more than a factor of 50. Without knowing details of how the HENRYWIN software calculates KH (this is mostly a black box) and upon which species the model was developed/trained, it is impossible to estimate the uncertainty in the calculated KH for an unknown species.

As the experimental values of KH_{diol} are not available in any published form, I suppose we will need to agree to disagree on the best way to estimate this quantity -- until this quantity has been measured experimentally. However, I think I make a logical argument that KH_{diol}(298K) of $1 \times 10^6 \text{ M atm}^{-1}$ is a plausible estimate. In this case, considering reasonable temperature dependencies (even to only 273K) pushes the KH_{diol} more than an order of magnitude larger than the value termed 'upper limit' in the paper for reduced T. This needs to be acknowledged.

We agree with the Referee's rationale. We now quote, in both the Methods (see below our answer to the next comment) and the SI Sect. 3.b.iii, that $\text{KH}(\text{diol}) = 10^6 \text{ M atm}^{-1}$ is also a possible estimate and that in that case higher solubility values could be reached too:

"However, if we assume that the solubility of methanediol at 298 K is similar to the one of hydroxymethyl hydroperoxide ($\sim 10^6 \text{ M atm}^{-1}$), methanediol solubility could be as high as $\sim 10^7 \text{ M atm}^{-1}$ at typical temperatures of warm clouds."

We have also changed a few statements to avoid presenting $\text{KH}(\text{diol}) = 10^4$ and 10^6 M atm^{-1} (and the model simulations) as strict lower and upper limits:

Lines 98-100: *"We gauge the impact of this uncertainty on the results by performing the simulations $\text{EMAC}_{(\text{diol})}$ and $\text{EMAC}_{(\text{dioh})}$ with a Henry's law constant (solubility) of $\sim 10^4$ and $\sim 10^6 \text{ M atm}^{-1}$, respectively (Methods)."*

Lines 133-134: *" $\text{EMAC}_{(\text{dioh})}$ and $\text{EMAC}_{(\text{diol})}$ provide, respectively, a lower and a higher estimate of the extra HCOOH produced via the multiphase processing of formaldehyde."*

Regarding the KH uncertainty, please see our response to the last comment.

LN 107 add 'at 298K' to end of sentence.

This has been added.

LN 434-436: I do not agree this statement. It can reasonably be estimated that $\text{KH}_{\text{diol}}(298\text{K}) = 1 \times 10^6 \text{ M/atm}$ (see last review), this along with reasonable temperature coefficient (see last review) yields $\text{KH}_{\text{diol}}(273\text{K}) = 1.4 \times 10^7 \text{ M/atm}$, and perhaps even higher KH at lower T, though as the authors suggest, things can become complicated as ice begins to form. As this is about 10x greater than upper estimate used, suggest this statement be modified.

We agree. In addition to the changes presented above, we have modified this statement as follows:

Lines 322-326: *"The values $\sim 10^4$ and $\sim 10^6 \text{ M atm}^{-1}$ are used for $\text{EMAC}_{(\text{diol})}$ and $\text{EMAC}_{(\text{dioh})}$, respectively. These are possible values of Henry's law constant for methanediol, given the spread of estimates at 298 K by semi-empirical methods and the expected temperature dependence. However, higher values of the order of 10^7 M atm^{-1} cannot be excluded at typical temperatures of warm clouds (see SI Sect. 3.b.iii)."*

SI Sect 1: It is not clear from the language here how the $\text{OH} + \text{HOCH}_2\text{OH}$ reaction coefficient was derived. Early on it says 7.5×10^{-12} was from fitting MCM simulations to data, while varying the $\text{OH} + \text{HOCH}_2\text{OH}$ rate coefficient in the simulations. Later, experimental rate coefficients of about 2×10^{-11} are written. Some more information regarding how the two relate and which is the best experimental estimate would be important to add. If the first part is written simply to justify the use of the number

in the global simulation, suggest eliminating this and simply state the global simulations were run before the experimental determination, but values are consistent, within uncertainties.

The rate constant of $7.5 \times 10^{-12} \text{ cm}^3 \text{ s}^{-1}$ is indeed from the best model fit to the experimental data (SI Sect. 1.b). Using a rate constant in the $1\text{-}10 \times 10^{-12} \text{ cm}^3 \text{ s}^{-1}$ range reproduces the observations within 5%. In SI Sect. 1.c, an independent determination of the rate constant yields values of $2.2(\pm 1.4)$ and $1.8(\pm 1.2) \times 10^{-11} \text{ cm}^3 \text{ s}^{-1}$ which are not significantly different from the model fit. In SI Sect. 2, very high-level theoretical kinetic predictions yield a rate constant of $\sim 1 \times 10^{-12} \text{ cm}^3 \text{ s}^{-1}$ with an estimated uncertainty of a factor of 3. Therefore, we have decided to perform the global simulations with the rate constant from the best fit to observations. This is now specified in SI Sect. 3.b.i as follows:

“The rate constant is from the best fit of the box model results to the SAPHIR experimental time series of formic acid and methanediol (see SI Sect. 1.b). This is consistent with the independent determination of the rate constant using the OH reactivity measurements (see SI Sect. 1.c), but outside the range of values from the theoretical predictions (see SI Sect. 2.b). A more accurate determination of this rate constant is needed.”

SI P18 last pp: As stated above, the ‘factor 3’ agreement of HENRYWIN KH with experimentally determined values is misleading. As stated in my first review, HENRYWIN KH for small glycols, arguably molecules much more similar to HOCH₂OH, can differ from experimental values by more than 50x. This statement needs to be corrected.

Indeed, the HENRYWIN’s estimate for 1,2-ethanediol differs from the observed KH by more than one order of magnitude. We have therefore adapted the statement in SI pp.18-19 as follows:

“For C₁-C₂ alcohols and hydroperoxides (including hydroxymethyl hydroperoxide), the HENRYWIN estimates with the bond method are within a factor of 3 of the experimental data, whereas for the small glycol 1,2-ethanediol, the estimate differs from the experimental values by more than an order of magnitude.”

Considering the HENRYWIN’s performance for these species, we assumed that the estimate for methanediol might be off by ~ 1 order of magnitude. This is now specified in SI Sect. 3.b.iii:

“This interval includes both an uncertainty on HENRYWIN predictions, set here at one order of magnitude, and the possible temperature dependence of the methanediol solubility, for which no constraint exists.”

Referee #4 (Remarks to the Author):

The authors have addressed my remaining comments in a satisfactory way. I recommend publication of the manuscript.

We thank the Referee for reviewing our study and for his/her positive evaluation.